# FUTUREFILL: FAST GENERATION FROM CONVOLUTIONAL SEQUENCE MODELS

**Naman Agarwal**[*]   **Xinyi Chen**[*]   **Evan Dogariu**[*]   **Devan Shah**[†]

**Hubert Strauss**[†]   **Vlad Feinberg**[*]   **Daniel Suo**[*]   **Peter Bartlett**[*]   **Elad Hazan**[*]

## ABSTRACT

We address the challenge of efficient auto-regressive generation in sequence prediction models by introducing FutureFill—a method for fast generation that applies to any sequence prediction algorithm based on convolutional operators. Our approach reduces the generation time from quadratic to quasilinear relative to the context length. Additionally, FutureFill requires a prefill cache sized only by the number of tokens generated, which is smaller than the cache requirements for standard convolutional and attention-based models. We validate our theoretical findings with experimental evidence demonstrating correctness and efficiency gains in language generation tasks. The project website can be found at dshah.io/futurefill.

## 1 INTRODUCTION

Large Transformer models (Vaswani et al., 2017) have become the method of choice for sequence prediction tasks such as language modeling and machine translation. Despite their success, they face a key computational challenge: the softmax attention mechanism incurs a quadratic computational cost during training and inference. This inefficiency has spurred interest in designing architectures that can handle long sequences more efficiently.

Convolution-based sequence prediction models (Li et al., 2022; Poli et al., 2023; Agarwal et al., 2023; Fu et al., 2024) have emerged as strong alternatives, primarily because they can leverage fast algorithms, in particular the Fast Fourier Transform (FFT) to achieve near-linear scaling in the sequence length during training. These models build on advances in State Space Models (SSMs), which have shown promise in modeling long sequences across diverse modalities (Gu et al., 2021a; Dao et al., 2022b; Gupta et al., 2022; Orvieto et al., 2023; Poli et al., 2023; Gu & Dao, 2023). Convolutional models offer a more general framework than SSMs because they can represent any linear dynamical system (LDS) without requiring parameters that scale with the dimensionality of the hidden states (Agarwal et al., 2023). This flexibility has led to recent developments that can handle longer contexts more effectively both in theory and practice. For instance, Spectral State Space Models or Spectral Transform Units (STUs) (Agarwal et al., 2023) use convolution-based spectral filtering algorithms (Hazan et al., 2017; 2018) to transform inputs into better-conditioned bases for long-term memory. The Hyena series of models (Poli et al., 2023; Massaroli et al., 2024) is another example, which learns implicitly parameterized Markov operators using convolution. Both methods exploit the duality between time-domain convolution and frequency-domain multiplication to accelerate prediction via the FFT algorithm.

While SSMs and recurrent models benefit from fast inference times independent of sequence length, convolutional models have significantly slower token generation times during inference. The best-known result for generating tokens with convolutional models is quadratic in sequence length—comparable to attention-based models (see Massaroli et al. (2024) Lemma 2.1). This limitation has prompted research into distilling SSMs from convolutional models (Massaroli et al., 2024), but the distilled SSMs are an approximation of the original convolutional models and the approximation gaps are not fully understood.

---

[*]Google DeepMind

[†]Princeton University

[*]Correspondence to: `namanagarwal@google.com`

In this paper, we consider exact auto-regressive generation from convolutional models, significantly reducing both the generation time and the cache size. We present our main results in two settings:

1. **Generation from Scratch:** When generating $L$ tokens from scratch, we demonstrate that long convolutional sequence predictors can generate these tokens in total time $O(L \log^2 L)$ with total memory $O(L)$. This improves upon previous methods that require $O(L^2)$ time for generation. We further provide a memory-efficient version wherein the total runtime increases to $O(L^{3/2}\sqrt{\log(L)})$ but the memory requirement is bounded by $O(\sqrt{L \log L})$.

2. **Generation with a Prompt:** When generating $K$ tokens starting from a prompt of length $L$, we show that the total generation time is $O(L \log L + K \log^2 K)$ with a cache size of $O(K)$. Previously, the best-known results for convolutional models were a total generation time bounded by $O(L \log L + LK + K^2)$ and a cache size bounded by $O(L)$ (Massaroli et al., 2024).

Importantly, our algorithms generate exactly from the convolutional model without relying on any approximations. There are numerous recent advances for efficient inference using approximate methods, for example cache compression (Zhang et al., 2024) and sparse attention (Beltagy et al., 2020). Since our approach involves no quality loss, we consider these methods to be in a different class and do not compare against them. Moreover, our methods are applicable to any convolutional model, regardless of how it was trained.

The following tables compares our algorithm with a standard implementation of convolution. It is worth noting that naive online convolution does not require additional memory beyond storing the inputs and filters. Our methods, however, provide a spectrum of trade-offs between computational complexity and memory usage. We also provide a comparison of the time and cache size requirements for exact computation in attention-based models.

| Method | Runtime | Memory |
|---|---|---|
| Standard Conv | $L^2$ | 1 |
| Standard Attn. | $L^2$ | 1 |
| EpochedFF (ours) | $L^{3/2}\sqrt{\log L}$ | $\sqrt{L \log L}$ |
| ContinuousFF (ours) | $L \log^2 L$ | $L$ |

| Prefill+Genertation Runtime | Generation Cache Size |
|---|---|
| $LK + L \log L + K^2$ | $L + K$ |
| $L^2 + KL$ | $L + K$ |
| $L \log L + K^{3/2}\sqrt{\log K}$ | $K$ |
| $L \log L + K \log^2 K$ | $K$ |

(a) Comparison for generating $L$ tokens from scratch. Runtime is in asymptotic notation, i.e. $O(\cdot)$ is omitted for brevity.

(b) Comparison for generating $K$ tokens starting from a prompt of length $L$, runtime and cache-size are in asymptotic notation.

To determine whether our theoretical findings lead to empirical benefits, we apply our algorithms to generate tokens in both controlled settings and from deep convolutional sequence prediction models. As a sanity check, we show empirically on isolated online convolutions that our algorithms achieve sub-quadratic scaling compared to the naive convolution implementation. We then consider more complex workloads where we generate from academic-sized models. We evaluate both purely convolutional and hybrid convolution/attention language models, and demonstrate that for both generating from scratch and generating from a prompt, our algorithms can achieve a substantial speedup of up to **1.7×** compared to the baseline.

## 1.1 RELATED WORK

Due to space limitations we provide a detailed related works section in the Appendix (Section D.1), and provide a short review in this section. Recurrent neural networks have been revisited in recent deep learning literature for sequence prediction in the form of state space models (SSMs), many of which can be parameterized as convolutional models. Gu et al. (2020) enable long-term memory via specialized system matrices, with follow-up works (Gu et al., 2021b;a; Gupta et al., 2022; Smith et al., 2023) improving stability and computational efficiency. Convolutional models such as LongConv (Fu et al., 2023), SGConv (Li et al., 2022), and Hyena (Poli et al., 2023) offer structured convolution kernel parameterizations for sequence prediction. For learning linear dynamical systems, spectral filtering (Hazan et al., 2017) emerges as a powerful, efficient method with provable

regret guarantees even in MIMO settings. This technique is developed under the online convex optimization (Hazan et al., 2016) framework, which lays the theoretical basis for adversarial sequence prediction. Given the strong guarantees, spectral filtering has been used to develop novel convolutional architectures for long range prediction and language modeling (Agarwal et al., 2023; Liu et al., 2024). Finally, independent work (Oncescu et al., 2024) presents a very similar algorithm for convolutional model inference with a total runtime of $O(L \log^2(L))$ (same as our Continuous-FutureFill result) using the method of relaxed polynomial interpolation. Our algorithms are based on the simple and intuitive idea of FutureFill, which allows us to create more practical algorithmic variants with lower memory usage and more streamlined implementation.

## 2 SETTING

**Notation:** For an input sequence $\{u_t\}$ we denote by $u_{1:t}$ the sequence of inputs $u_1, ..., u_t$. For any $i \leq j$ let $u_{i:j}$ denote the sub-sequence $u_i, u_{i+1}, \ldots u_j$. When $i > j$, $u_{i:j}$ denotes the subsequence $u_{j:i}$ in reverse order. We also denote $[k] = \{1, 2, ..., k\}$ as a set of $k$ natural numbers. For a vector $u$, let $[u]_j$ denote the $j$-th coordinate of $u$; if $u$ is a one-dimensional sequence, then let $[u]_j$ denote the $j$-th position of $u$. Given a multi-dimensional sequence $u_1 \ldots u_t$ where each $u_i \in \mathbb{R}^d$ and given a vector $v \in \mathbb{R}^t$, for brevity we overload the definition of inner products by defining $y = \langle v, u_{1:t} \rangle$ with $y \in \mathbb{R}^d$ as $y_j = \sum_{i=1}^{t} v_i \cdot [u_i]_j \in \mathbb{R}$. That is, $y$ is a $d$-dimensional vector where the coordinate $j$ is the inner product between $v$ and the sequence $[u_1]_j, \ldots, [u_t]_j$.

**Convolution:** The convolution operator between two vectors $u, \phi \in \mathbb{R}^t$ outputs a sequence of length $t$ whose element at any position $s \in [t]$ [1] is defined as

$$[u * \phi](s) = \sum_{i=1}^{s} u_i \phi_{s+1-i} = \langle u_{1:s}, \phi_{s:1} \rangle. \tag{1}$$

A classical result in the theory of algorithms is that given two vectors $u, \phi \in \mathbb{R}^t$, their convolution can be computed in time $O(t \log t)$, using the FFT algorithm.

**Online Convolution:** We consider the problem of performing the convolution $u * \phi$ when one of the sequences $\phi$ is fully available to the algorithm, however the other sequence $u$ *streams* in – the element $u_t$ is made available to the algorithm at the start of round $t$, at which point it has to release the output $[u * \phi]_t$. This model of online convolution is immediately relevant to the online auto-regressive generation of tokens from a convolutional sequence model, as the output token at time $t$ becomes the input for the next round. In this setting, the sequence $u$ corresponds to generated tokens and the sequence $\phi$ corresponds to the convolutional filter which is known to the model. We further detail the setup of sequence generation in the next subsection.

**Naive Online Convolution:** Online convolution can be implemented by directly computing the inner product at each time step, as the new input becomes available. We refer to this method as naive online convolution. It has a computational complexity of $O(L^2)$ for predicting for $L$ steps and requires no additional memory beyond storing the inputs and filters.

### 2.1 AUTO-REGRESSIVE SEQUENCE PREDICTION

**Sequence Prediction:** In this setting, the input is a sequence of tokens denoted $u_1, ..., u_t, ...$, where $u_t \in \mathbb{R}^{d_{in}}$. The predictor's task is to generate a sequence $\hat{y}_1, ..., \hat{y}_t, ...$, where $\hat{y}_t \in \mathbb{R}^{d_{out}}$ is generated after observing the inputs $u_1, ..., u_{t-1}$. The output $y_t$ is observed after the predictor generates $\hat{y}_t$. The quality of the prediction is measured by the distance between the predicted and observed outputs according to a loss function $\ell_t(\hat{y}_t, y_t)$, for example the $\ell_2$ distance $\|\hat{y}_t - y_t\|^2$.

**Auto-regressive Sequence Prediction:** When predicting a sequence in an auto-regressive fashion, in each iteration an online predictor first makes a prediction using the existing inputs $u_1, \ldots, u_{t-1}$, and then append the prediction $\hat{y}_t$ to the inputs to be used in the next iteration, where the inputs

---

[1] This definition corresponds to the *valid* mode of convolution in typical implementations of convolution e.g. scipy.

become $u_1, \ldots, u_{t-1}, \hat{y}_t$. When predicting from scratch, the online predictor starts from a given initial token and predicts, or generates, the rest of the sequence.

**Auto-regressive Sequence Prediction from a Prompt:** Auto-regressive sequence prediction starting from a prompt is commonly used by large language models. Herein the sequence model has to generate a specified number of tokens given a certain context. In practice, this setting consists of two stages, the prefill stage and the decode stage.

During prefill, the model ingests the entire context and generates a cache that stores context information required for generation. When decoding, the model takes the cache and the most recently generated token as input and generates the next output token. The cache is then updated with the most recent input token. The cache stores the input information the prediction algorithm needs in order to generate the output. For instance, Transformers typically save the key and value vectors of past inputs in a KV cache, and for convolutional models, naive online convolution stores all previous inputs. As a result, for these models, generating $K$ tokens from a prefill of length $L$ requires a cache of size $O(L+K)$. This can be prohibitively large for long-context inference with an extensive prompt, and reducing the cache size is key in this setting (Hooper et al., 2024).

### 2.2 Online Convolutions in Sequence Prediction

We define a convolutional sequence prediction model to be given by a *filter*, which is a vector denoted by $\phi \in \mathbb{R}^L$ where $L$ is the *context length* of the model. It takes as an input a sequence $u$, and outputs a prediction at time $t$ according to the following equation, $\hat{y}_t = \langle \phi, u_{t:t-L} \rangle$.

The above definition can be extended to include nonlinearities and multiple filter *channels*. This paradigm captures several prominent convolutional sequence models considered in the literature, and we highlight some of them below (additional details are provided in the appendix in Section D.2). Our online convolution techniques can be straightforwardly applied to all the following models, leading to an improvement in the generation time from $O(L^2)$ to $\tilde{O}(L)$. When generating from a prompt, we improve the cache size from $O(L + K)$ to $O(K)$.

**Spectral Transform Units:** The STU architecture was proposed in (Agarwal et al., 2023) based on the spectral filtering technique for linear dynamical systems (Hazan et al., 2017; 2018). These are convolutional sequence models based on carefully constructed filters that are **not data-dependent**. More specifically, the filters $\phi_1, ..., \phi_k$ are derived from a fixed Hankel matrix $H_L$ depending only on the sequence length $L$. The STU predicts according to the following rule [2] $\hat{y}_t = \sum_{i=1}^{k} M_i \langle \phi_i, u_{t:t-L} \rangle$, where $M_{1:k}$ are learned projection matrices. Note that the inner products $\langle \phi_i, u_{t:t-L} \rangle$ are the outputs of $\phi_i * u$. The STU architecture is particularly appealing for learning LDS with long memory, as demonstrated by its dimension-free sublinear regret guarantees for this setting (Agarwal et al., 2023). For more details see Appendix D.2.

**Hyena:** The Hyena architecture proposed in Poli et al. (2023) sequentially applies convolutions and element-wise products in an alternatve fashion. Formally, given an input $u_{1:t}$, $N + 1$ linear projections $v, x_1, \ldots x_N$ of the input are constructed (similar to the $q, k, v$ sequences in self-attention). The hyena operator as a sequence of convolutions with learnable filters $h_1 \ldots h_N$ is then given by

$$y = x_N \cdot (h_N * (x_{N-1} \cdot (h_{N-1} * (\ldots)))) .$$

## 3 Efficient Online Convolutions using FutureFill

We begin by introducing a simple and convenient primitive named FutureFill that forms the crucial building block of our algorithms. Intuitively, FutureFill corresponds to computing the *contribution* of the current and previously generated tokens on the future tokens yet to be generated. For a convolutional model (and unlike attention) this contribution can be efficiently determined without even having generated the future tokens. Here onwards, for brevity of notation, for any $v \in \mathbb{R}^t$, we assume $v_j = 0$ for any $j \leq 0$ or any $j > t$. Formally, given two sequences $v \in \mathbb{R}^{t_1}, w \in \mathbb{R}^{t_2}$ we

---

[2]more precisely, there are additional linear and constant terms depending on the exact filters used, such as $\hat{y}_t = \hat{y}_{t-2} + \sum_{i=1}^{3} M_i^u u_{t+1-i} + \sum_{i=1}^{k} M_i \langle \phi_i, u_{t:t-L} \rangle$, see Agarwal et al. (2023) for more details.

define $\text{FutureFill}(v, w) \in \mathbb{R}^{t_2-1}$ as [3]

$$\forall s \in [t_2 - 1] \quad [\text{FutureFill}(v, w)]_s = \sum_{i=1}^{t_2-s} v_{t_1-i+1} \cdot w_{s+i}.$$

Figure 5 in Appendix D.3 depicts the FutureFill operation between an input sequence and a convolutional filter. Conceptually, $[\text{FutureFill}(v, w)]_s$ is the contribution of the input $v$ of length $t_1$ to the output $[v * w]$ at position $t_1 + s$. The FFT algorithm for convolutions can easily be extended to compute the FutureFill as well in time at most $O((t_1 + t_2) \log(t_1 + t_2))$. For example, the *full* mode of a standard conv implementation (e.g., numpy) can be used to compute FutureFill in the following way under Python slicing convention (exclusive of the last index),

```
FutureFill(v, w) = numpy.convolve(v, w, mode=full)[t_1:t_1+t_2-1]
```

To leverage FutureFill for efficient generation from a convolutional model, consider the proposition below that follows from the definition of convolution.

**Proposition 1.** *Given two vectors $a, b \in \mathbb{R}^t$, we have that $\forall t_1, s \in [t]$,*

$$[a * b]_s = \begin{cases} [a_{1:t_1} * b_{1:t_1}]_s & \text{if } s \le t_1 \\ [a_{t_1+1:t} * b_{1:t-t_1}]_{s-t_1} + [\text{FutureFill}(a_{1:t_1}, b)]_{s-t_1} \end{cases}$$

That is, the convolution of two vectors $a$ and $b$ can be broken into a FutureFill operation and another convolution involving $b$ and only the most recent positions of $a$. We provide a proof in the appendix.

### 3.1 Epoched-FutureFill: Efficient Online Convolution

When computing online convolutions, the FutureFill routine efficiently pre-computes the effect of past tokens on future ones. We leverage this property in the Epoched-FutureFill procedure outlined in Algorithm 1 to compute online convolutions.

---

**Algorithm 1** Epoched-FutureFill: Efficient Online Convolutional Prediction

---

1: **Input:** Filter $\phi \in \mathbb{R}^L$. Input seq. $u \in \mathbb{R}^L$, streaming coordinate-wise. $K$, the epoch length.
2: Set $\tau = 1$. Set FutureFill cache $C \in \mathbb{R}^K$ to 0.
3: **for** $t = 1, 2, ..., L$ **do**
4:     Receive $u_t$, and compute and output $\hat{y}_t = \sum_{j=1}^{\tau} u_{t+1-j} \cdot \phi_j + C_\tau$.
5:     **if** $\tau = K$ **then**
6:         Compute FutureFill cache $C \in \mathbb{R}^K$ defined as $C_j = [\text{FutureFill}(u_{1:t}, \phi_{1:t+K})]_j$.
7:         $\tau \leftarrow 1$
8:     **else**
9:         $\tau \leftarrow \tau + 1$
10:     **end if**
11: **end for**

---

The following theorem establishes the properties of Epoched-FutureFill and provide a trade-off between the additional memory overhead and total runtime incurred by the algorithm. In particular, the runtime in this trade-off is optimized when the total memory is $O(\sqrt{L \log L})$, leading to a total runtime of $O(L^{3/2}\sqrt{\log L})$.

**Theorem 2.** *Algorithm 1 computes the online convolution of sequences with length $L$ and runs in total time $O\left(\frac{L^2 \log L}{K} + KL\right)$ with a total additional memory requirement of $O(K)$. Setting $K = \sqrt{L \log L}$ to minimize the runtime, Algorithm 1 computes online convolution in $O(L^{3/2}\sqrt{\log L})$ total time and $O(\sqrt{L \log L})$ memory.*

*Proof.* Since the proof of correctness is mainly careful accounting of various terms, we provide it in the appendix and give the running time results in this proof. The running time consists of two

---

[3]recall that we denote $[x] = \{1 \dots x\}$.

components. First, at every iteration, line 4 is executed. One term, $C_\tau$, has already been computed and saved in line 6, so we can retrieve it in constant time. The other term is a sum of $\tau$ products, which can be computed in time $O(\tau)$. Second, every $K$ iterations, we execute line 6 and update the cache. The FutureFill operation can be computed via the FFT in at most $O(L \log L)$ time.

Summing over $L$ iterations, the total computational complexity is

$$\frac{L}{K} \left( L \log L + \sum_{\tau=1}^{K} \tau \right) = O\left( \frac{L^2 \log L}{K} + KL \right) = O\left( L^{3/2} \sqrt{\log L} \right),$$

where the last equality holds when the cache size $K = \sqrt{L \log L}$ is chosen to minimize the sum. $\quad\square$

## 3.2 CONTINUOUS-FUTUREFILL: QUASILINEAR ONLINE CONVOLUTION

In this section we specify a procedure that significantly improves upon the runtime of Epoched-FutureFill. Our starting point is Proposition 1, which implies that to compute the convolution between two sequences, we can break the sequences at any point, compute the convolution between the corresponding parts and *stitch* them together via a FutureFill computation. This motivates the following Divide and Conquer algorithm to compute the convolution of two sequences $a, b \in \mathbb{R}^L$

- Recursively compute $a_{1:L/2} * b_{1:L/2}$, $a_{L/2+1:t} * b_{1:L/2}$.
- Output the concatenation of $a_{1:L/2} * b_{1:L/2}$ and $(a_{L/2+1:t} * b_{1:L/2}) + \text{FutureFill}(a_{1:L/2}, b)$.

Since FutureFill for $L$-length sequences can be computed in time $O(L \log L)$ via the FFT, a standard divide-and-conquer approach yields an $O(L \log^2 L)$ computational complexity for the algorithm. Although this complexity is worse than an FFT, the advantage of the above method is that it can be executed *online*, i.e. the tokens can be generated as input streams in.

We provide a formal description of the algorithm in Algorithm 2. We note that the algorithm description essentially serializes the sequence of operations involved in the above divide-and-conquer procedure by their chronological order. For high-level intuition, we encourage the reader to maintain the divide-and-conquer structure when understanding the algorithm. The algorithm proceeds as follows: at each time step, $\hat{y}_t = \langle u_{1:t}, \phi_{t:1} \rangle$ is returned as a sum of $C_t$, the cache that stores the contribution from past tokens, and $u_t \cdot \phi_1$, the contribution from token $u_t$. In Line 7, the algorithm then computes the contribution of tokens $u_{t-2^{k(t)}+1:t}$ to positions $t+1, \ldots, t+2^{k(t)}$ of $[u * \phi]$. Finally, we add the output of FutureFill to the existing cache $C$ to accumulate the contributions. We provide a schematic illustrating the flow of the algorithm in the Appendix (Section D.3). In the following theorem we provide a running time bound for Algorithm 2 and defer the proof to the Appendix (Section F).

**Theorem 3.** *Algorithm 2 computes the online convolution of sequences with length $L$ and runs in total time $O(L \log^2(L))$ with a total additional memory requirement of $O(L)$.*

---

**Algorithm 2** Continuous-FutureFill: Quasilinear Generation From Convolutional Models

---

1: **Input:** Convolutional filter $\phi \in \mathbb{R}^L$. Input sequence $u \in \mathbb{R}^L$, streaming one coordinate every round.
2: Set $b = \lfloor \log L \rfloor$. Set FutureFill cache $C \in \mathbb{R}^L$ to 0.
3: **for** $t = 1 \ldots L$ **do**
4:     Receive $u_t$. Output $\hat{y}_t = C_t + u_t \cdot \phi_1$.
5:     Let $k(t)$ be the highest power of 2 that divides $t$, i.e. $k = \max\{i \in [b] : t \mod 2^i = 0\}$.
6:     Compute $\text{FF} = \text{FutureFill}(u_{t-2^{k(t)}+1:t}, \phi_{1:2^{k(t)+1}})$
7:     Set $C_i = C_i + \text{FF}_{i-t} \quad \forall \ i \in [t+1, t+2^{k(t)}]$
8: **end for**

---

**Remark 4.** The computational complexity of Continuous-FutureFill is optimal up to poly-log factors as we cannot hope to generate faster than constant time per token, and further improvements remain an open question.

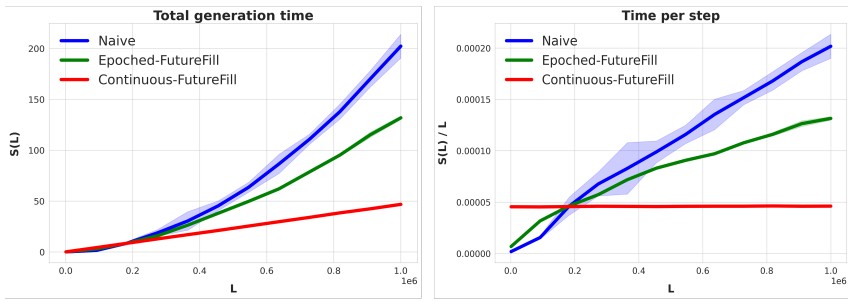

Figure 1: Total and average number of seconds per step when generating $L$ tokens.

## 4 FAST AUTO-REGRESSIVE SEQUENCE GENERATION FROM A PROMPT

In this section we consider the problem of auto-regressively generating $K$ tokens starting from a given prompt of length $L$. For convolutional models in particular, we define an abstract version of the problem: given a prompt vector $p \in \mathbb{R}^L$ and a convolutional filter $\phi \in \mathbb{R}^{L+K}$ [4], the aim is to iteratively generate the following sequence of tokens

$$\hat{y}_t = \langle \hat{y}_{1:t-1}, \phi_{t-1:1} \rangle + \langle p_{1:L}, \phi_{t+L-1:t} \rangle = \sum_{j=1}^{t-1} \hat{y}_{t-j} \cdot \phi_j + \sum_{j=t}^{t+L-1} p_{t+L-j} \phi_j.$$

As the above definition clearly shows, the expected output is an online convolution where the input sequence $u$ has a prefix of the prompt $p$ and the input sequence is appended by the most recently generated output by the model (i.e. auto-regressive generation). Observe that the output can be computed from a FutureFill operation and another online convolution involving the generated tokens, which can be computed using either of our online convolution algorithms. In the Appendix (Section D.4), we formally provide Algorithm 3 that specifies the above method using Continuous-FutureFill (Algorithm 2) as the online convolution algorithm. The corollary below bounds the running time for the overall method which follows easily from Theorem 3.

**Corollary 5.** *Algorithm 3 when supplied with a prompt of sequence length $L$, generates $K$ tokens in total time $O(L \log L + K \log^2 K)$ using a total cache of size $O(K)$.*

## 5 EXPERIMENTS

### 5.1 CONTROLLED SETTINGS

In this section, we verify our theoretical results on isolated online convolution operations. We randomly initialize one-dimensional filters and study the setting where we generate from scratch, where algorithms generate $L$ outputs from a given initial input. We evaluate Epoched-FutureFill (Algorithm 1) which has a runtime of $O(L^{3/2}\sqrt{\log L})$ and Continuous-FutureFill (Algorithm 2) which has a runtime of $O(L \log^2 L)$ against the naive implementation, which has a runtime of $O(L^2)$. For increasing values of $L$, we measure the time $S(L)$ it takes to generate $L$ outputs. In Figure 1 we plot the amortized step time $S(L)/L$ and total generation time $S(L)$, respectively, as functions of $L$. The behavior is consistent with our theory: the naive algorithm runs in amortized $O(L)$ per step, while our methods achieve sublinear and logarithmic runtime complexities respectively. In the appendix (Section C) we present additional experiments where we show that Epoched-FutureFill significantly outperforms Transformer models with a standard KV cache and convolutional models with naive decoding (the state of the art for convolutional models) for inference.

---

[4]The assumption of the filter being larger than $L + K$ is without loss of generality as it can be padded with 0s

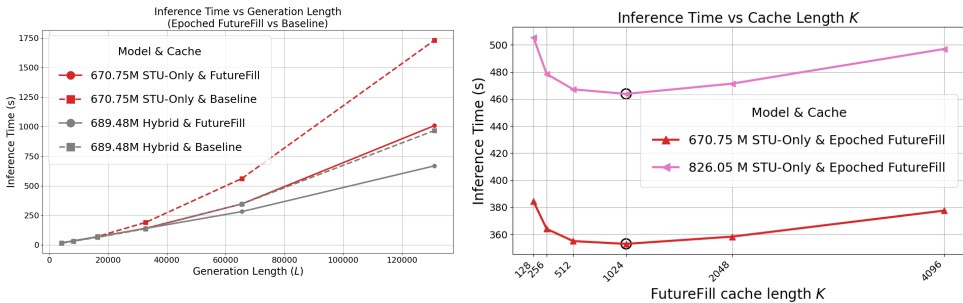

(a) Decode time (in s), without prefill. Baselines are in dashed lines.

(b) Decode time (in s) for a fixed generation length of 65,536 tokens without prefill.

Figure 2: Decode time without prefill and ablations on cache length K

## 5.2 EXPERIMENTS ON CONVOLUTIONAL LANGUAGE MODELS

In this section, we further show that our theoretical results on FutureFill's sub-quadratic generation time hold when using academic-sized convolutional language models, i.e. models of up to 826.05M parameters. We focus here on the more practical Epoched-FutureFill (Algorithm 1).

### 5.2.1 SETUP

We conduct our experiments using two variants of FlashSTU-T, a convolutional model based on Spectral Transform Units as introduced in Liu et al. (2024):

- **Fully convolutional variant**: This model consists entirely of Spectral Transform Units (STUs) with the tensordot approximation, which convolve the projected input against fixed spectral filters and apply an MLP layer. End-to-end default precision is `float32`.

- **Hybrid variant**: This model combines STU blocks with local attention blocks at 1:1 ratio. We adhere closely to the setup specified in Liu et al. (2024), apart from the number of layers and input dimensions as those will be modified in our ablations and the number of attention heads which is set to 4. See Appendix B.1 for more details. End-to-end default precision is `bfloat16`; only the FFT operation is upcasted to `float32`.

Since our focus in the ablations below is primarily on generation speed rather than downstream performance, we initialize the filters ($\phi_{1:k}$) uniformly at random while following the initialization approach detailed in Liu et al. (2024) for all other layers. To validate correctness, we also generate text from a 330M pretrained FlashSTU-T model and evaluate on downstream tasks. Our results show that the accuracy matches that of a baseline using naive online convolution (Appendix B.6).

Without prefill, we conduct three ablations—generation length, model depth, and model width—starting each run from the `<|endoftext|>` token with no prefill cache. With prefill, we ablate the input prompt length $L_{prompt}$, generation length $L_{gen}$, and the depth and width of the fully convolutional variant of FlashSTU-T. Additionally, we ablate the cache size $K$, which we previously set to the theoretical optimum $K = \sqrt{L_{\text{gen}} \log L_{\text{gen}}}$ (Theorem 2). More details on the ablation experiments are available in Appendix A.

### 5.2.2 RESULTS

All experiments were run on a single NVIDIA H100 GPU. For each model configuration and sequence length, we measure the total decode time (including the full forward pass through MLPs and STU/attention blocks) over three successive runs and report the average of the final two.

**Ablations Without Prefill** Figure 2a reports the decode times for our two largest models (12 layers, 1024-dim input, 4 attention heads when applicable): the 670.75M-parameter STU-only model

and the 689.48M-parameter hybrid model. Across all generation lengths, Epoched-FutureFill exhibits clear sub-quadratic scaling, while the baseline shows near-quadratic growth in runtime.

Indeed, as $L$ grows larger, the runtime advantage of Epoched-FutureFill becomes more noticeable. At the largest generation length $L_{gen} = 126{,}976$, we observe a **1.7× speedup** for STU-only and a **1.5× speedup** for the hybrid variant, compared to a naive convolution (baseline). More results for different combinations of depth and width are provided in Appendix B.2 and we achieve consistent speedup.

**Ablations With Prefill, Input Prompt Longer than Generation**  In the case of prefilling, we measure the prefill time separately from the decode time. We average the prefill time per model over the generation length.

| Parameter count | Input dim | Layer count | Cache Type | Avg Prefill Time (s) | Decode time (s) at generation length $L_{\text{gen}}$ | | |
|---|---|---|---|---|---|---|---|
| | | | | | 4096 | 8192 | 16384 |
| 515.46M (STU only) | 1024 | 8 | Epoched FutureFill | 21.40 | $13.12 \pm 0.05$ | $26.18 \pm 0.01$ | $52.22 \pm 0.08$ |
| 515.46M (STU only) | 1024 | 8 | Baseline | 21.28 | 25.23 | $52.31 \pm 0.02$ | $111.92 \pm 0.06$ |
| 670.75M (STU only) | 1024 | 12 | Epoched FutureFill | 31.98 | $19.06 \pm 0.1$ | $37.80 \pm 0.01$ | $75.66 \pm 0.61$ |
| 670.75M (STU only) | 1024 | 12 | Baseline | 37.20 | $37.21 \pm 0.02$ | $77.15 \pm 0.01$ | $165.13 \pm 0.07$ |

Table 2: End-to-End decode time (in s) with prefill on an input prompt of length $L_{\text{prompt}} = 32{,}768$ tokens. Error bars are $\pm 1$ sample standard deviation over the two post–warmup runs.

Figure 2 reports the decode and prefill times in seconds for our two largest models (12 layers, 1024-dim input, 4 attention heads when applicable): the 670.75M-parameter STU-only model and the 689.48M-parameter hybrid model. Epoched-FutureFill's decoding is substantially faster with increasing model size. It is even noticeable for smaller generation length when the initial prefill is large, as shown in Table 2, because the naive baseline recomputes the full prompt convolution at every token. At the largest generation length $L_{\text{gen}} = 16{,}384$, for a prefill length of 32,768 tokens, we observe a **2× speedup** for both models, compared to the naive cached convolutions (baseline).

For further examples of depth–width pairings in the case where the input prompt exceeds the generation length, see Appendix B.3.1. In the less common scenario - when the generation length is far longer than the input prompt - Appendix B.3.2 presents ablations over depth, width and generation length. In every case, the observed speedups are robust and grow steadily as the model scales.

To arrive at a more fine-grained understanding of the speedups, we provide module-wise timing breakdowns for the 670.75M-parameter STU-only model in Appendix B.5. Our results show that convolution is by far the most time-consuming component of the model. These results further highlight that Epoched-FutureFill significantly decreases the STU (conv/FFT) part of end-to-end decoding time to achieve the observed speedups, while other components change only marginally.

**Remark 6.** In our implementation, we computed the prefill pass on the same GPU that performed decoding, occasionally triggering GPU OOM errors when using very long prompts. In practical deployments, the prefill cache is typically produced on a separate host and then loaded onto the decoding GPU, eliminating this memory bottleneck.

**Ablations on $K$, without prefill**  When the generation length is set to 65,536 tokens, the optimal $K = \sqrt{L_{\text{gen}} \log L_{\text{gen}}}$ is 1024 according to Theorem 2. We empirically verify it in Figure 2b: there is a monotonic improvement in the decode time until the theoretically optimal value of $K$.

**Precision of the FFT & Downstream Tasks**  Using lower precision is common in language model inference. To understand its effect on our algorithm, we compare upcasting the FFT to fp32 vs fp16 while keeping the model in bf16. We chose fp16 because PyTorch FFT does not support bf16 and lower precisions, while upcasting to fp32 is consistent with the method we used to measure generation speed. Our results show that the lower precision does not have a significant impact. Accuracy matches the naive convolution baseline in bf16 across WinoGrande, HellaSwag, and PiQA, using an openly-available 330M-parameter pretrained hybrid FlashSTU-T (Appendix B.6).

## 6 CONCLUSIONS

In this paper, we address the problem of efficiently generating from convolutional models and present FutureFill, a novel method that reduces the computational complexity of generating $L$ tokens from $O(L^2)$ to $O(L \log^2 L)$. We introduce a simple but powerful subroutine which enables a flexible runtime/memory trade-off, making our method adaptable to different practical settings. Our experiments confirm the theoretical improvements, demonstrating significant efficiency gains in convolution sequence model generation. These results suggest that FutureFill can serve as an efficient alternative to existing methods, particularly for problems requiring long-sequence modeling.

**Reproducibility Statement:** We have made careful efforts to aid reproducibility of our work and ensure that our results, and limitations of our results, are well understood. In the Appendix, we have provided both an overview of each experiment and granular implementation details to aid in reproducibility. We have additionally uploaded our modification of FlashSTU (Liu et al., 2024) with the FutureFill algorithm as a supplemental file and provide full details of our experimental setup. Beyond this, we devote much of the appendix to providing full proofs and grounding for our theoretical arguments and to offering clarifying case studies. We are excited about the potential impact of FutureFill and care deeply about ensuring our work is in the best possible position for adoption.

## 7 ACKNOWLEDGEMENTS

The authors thank Annie Marsden for useful discussions though the development of the project. EH gratefully acknowledges support from the Office of Naval Research and Open Philanthropy.

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

## A    EXPANDED ABLATION DETAILS

**Ablations Without Prefill**: We conduct three ablations—generation length, model depth, and model width—starting each run from the `<|endoftext|>` token with no prefill cache. Specifically, we:

- Vary generation length $L_{\text{gen}}$ from $4096 \rightarrow 126976$.
- Vary the depth of the model within $[6, 8, 12]$ layers while keeping a fixed input dimension of $1024$.
- Vary the width of the model within $[512, 896, 1024]$ while keeping a fixed number of layers at $12$.

**Ablations with Prefill**: In this set of ablations, we only consider the fully convolutional variant of FlashSTU-T. Generation is initiated from a prompt which we use in the prefill stage. We:

- Vary the length of the input prompt $L_{prompt}$ from $512 \rightarrow 32768$. The length of the generation $L_{\text{gen}}$ is varied from $4096 \rightarrow 130560$.
- Vary the depth and width of the model in the same manner as in the ablations without prefill, and fix the input dimension to be $1024$, the number of layers to be $12$.

In both sets of ablations, our baseline for comparison is naive online convolution, where past activations are stored and the convolution is recomputed across the entire sequence at each generation step. These configurations yield models ranging from 160.71M to 689.48M parameters. Refer to Appendices B.2 and Appendix B.3 for more details.

**Ablation on cache size $K$**   : Across all our previous experiments, we set the FutureFill epoch length to the theoretical optimum $K = \sqrt{L_{\text{gen}} \log L_{\text{gen}}}$ (Theorem 2). To abate this choice, we consider a fixed generation length $L_{\text{gen}}$ of $65536$ tokens without prefill, and we sweep $K$ using fully convolutional FlashSTU-T of various sizes between 417.08 M and 826.05M.

## B    ADDITIONAL IMPLEMENTATION DETAILS AND ABLATIONS

### B.1    ADDITIONAL IMPLEMENTATION DETAILS

All experiments were run on a single NVIDIA H100 GPU. All timings were measured over three independent runs. For each configuration, we discard the first run and compute the mean and sample standard deviation over the remaining two. Error bars represent these sample standard deviations and are omitted whenever the standard deviation is below 0.01 s. Appendix timing tables report wall-clock decode time in seconds; throughput can be calculated via $L_{gen}/T$.

We employ the FlashSTU-T architecture from Liu et al. (2024). Our ablations use either a *hybrid* variant—alternating between STU-T blocks and sliding-window attention layers—or a fully convolutional *STU-T only* variant. Inputs are tokenized with the o200k_base tokenizer and embedded with tied weights between the input embedding and output unembedding matrices. We add special tokens (`<|endofprompt|>`, `<|endoftext|>`) to signal generation boundaries.

For our attention layers, we leverage FlashAttention v2 (Dao et al., 2022a; Dao, 2024) with ALiBi positional encodings (Press et al., 2022). Each MLP layer has a hidden dimension $12\times$ the input dimension.

Let $x_1, x_2, \ldots, x_\ell \in \mathbb{R}^d$ be the inputs the Spectral Transform Unit (STU) layer. the STU leverages $k = 48$ spectral filters $\phi_1, \ldots, \phi_k \in \mathbb{R}^L$, with $L \geq \ell$, to compute $U_j = \sum_{i=0}^{\ell-1} x_{\ell-i} \cdot \phi_j(i) \in \mathbb{R}^d$. The STU maintains learned parameters $M_j \in \mathbb{R}^{d \times d}$ to compute output $x_{\ell+1} = \sum_{j=1}^{k} M_j U_j$. Thus, the STU layer involves $k \cdot d$ convolutions per auto-regressive generation. The STU-T represents the Spectral Transform Unit with the tensordot approximation as introduced in Liu et al. (2024). For an STU with tensordot approximation, rather than maintaining $k$ matrices $M_j \in \mathbb{R}^{d \times d}$, concatenated as $M \in \mathbb{R}^{d \times k \times d}$, we approximate $M \approx M^{(1)} \times M^{(2)}$, with $M^{(1)} \in \mathbb{R}^{k \times d}$ and $M^{(2)} \in \mathbb{R}^{d \times d}$. This allows for a more convenient computation, as we can compute $x_{\ell+1} = \sum_{i=0}^{\ell-1} (M^{(2)} x_{\ell-i}) \odot M_{\text{filters}}[i]$, where $M_{\text{filters}} = [\phi_1, \ldots, \phi_k]^\top M^{(1)} \in \mathbb{R}^{L \times d}$ and $\odot$ denotes element-wise product. Thus, the STU with tensordot approximation requires only $d$ convolutions per token. Although the STU

performance guarantees on learning linear dynamical systems (Appendix D.2) no longer holds for the tensordot approximation, the STU-T retains practical performance as shown in Liu et al. (2024).

Convolutions within each STU-T block are implemented via an FFT-based operation: given batched inputs $v \in \mathbb{R}^{B \times t_1}$ and filters $w \in \mathbb{R}^{B \times t_2}$, we zero-pad both to the next power-of-two length $n_{\text{FFT}}$ when memory allows, perform real-valued FFTs (rfft), multiply pointwise in the frequency domain, and invert back with irfft. This yields the causal convolution output in $\mathcal{O}(n_{\text{FFT}} \log n_{\text{FFT}})$ time, which is significantly faster than direct convolution for long sequences. Note that, for memory reasons, we do this padding to the next power-of-two only for FFT sizes below 131072. In our PyTorch implementation (Paszke et al., 2019), we cast inputs to float32 for FFT compatibility, and finally truncate the result to the causal FutureFill window before casting back to the original dtype.

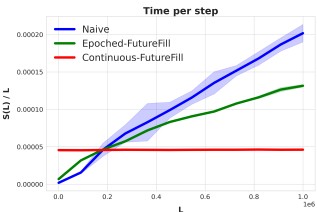

Figure 3: FlashSTU-T architecture. Figure from Liu et al. (2024).

**FlashSTU** Liu et al. (2024) source code is publicly available at https://github.com/hazan-lab/flash-stu. It is released under the Apache License, Version 2.0, which permits unrestricted use, modification, and distribution with attribution.

**PyTorch** (Paszke et al., 2019) is used for the implementations and associated experiments, in the current section 5.2 of the main paper and section B of the Appendix. PyTorch's code is hosted at https://github.com/pytorch/pytorch (tag v2.0.0) and distributed under the BSD 3-Clause License, allowing use and redistribution with minimal restrictions.

**FlashAttention** (Dao et al., 2022a; Dao, 2024) is publicly available at https://github.com/Dao-AILab/flash-attention. As is PyTorch, FlashAttention is distributed under a BSD 3-Clause License, allowing use and redistribution with minimal restrictions.

## B.2 ADDITIONAL ABLATIONS, WITHOUT PREFILL

| Parameter count | Input dim | Layer count | Cache Type | Decode time at generation length $L_{gen}$ | | | | | |
|---|---|---|---|---|---|---|---|---|---|
| | | | | 4096 | 8192 | 16384 | 32768 | 65536 | 126976 |
| 160.71M | 512 | 6 | Epoched FutureFill | $9.31 \pm 0.01$ | $18.42 \pm 0.03$ | $37.08 \pm 0.05$ | $74.05 \pm 0.30$ | $147.74 \pm 0.15$ | $322.28 \pm 0.06$ |
| 180.13M | 512 | 8 | Epoched FutureFill | $11.85 \pm 0.01$ | $23.60 \pm 0.05$ | $46.87 \pm 0.03$ | $93.29 \pm 0.08$ | $187.12 \pm 0.55$ | $416.87 \pm 0.15$ |
| 218.98M | 512 | 12 | Epoched FutureFill | $17.02 \pm 0.07$ | $33.85 \pm 0.02$ | $68.20 \pm 0.09$ | $135.70 \pm 0.33$ | $272.35 \pm 0.21$ | $611.70 \pm 0.57$ |
| 357.62M | 896 | 6 | Epoched FutureFill | $9.35 \pm 0.01$ | $18.60 \pm 0.04$ | $37.33 \pm 0.07$ | $74.70 \pm 0.15$ | $170.16 \pm 0.01$ | $468.34 \pm 0.04$ |
| 417.08M | 896 | 8 | Epoched FutureFill | $11.77 \pm 0.01$ | $23.58 \pm 0.03$ | $47.12 \pm 0.05$ | $94.03 \pm 0.71$ | $219.96 \pm 0.20$ | $613.40 \pm 0.09$ |
| 535.99M | 896 | 12 | Epoched FutureFill | $17.06 \pm 0.04$ | $34.12 \pm 0.02$ | $67.98 \pm 0.12$ | $137.01 \pm 0.20$ | $321.47 \pm 0.08$ | $904.04 \pm 0.28$ |
| 437.81M | 1024 | 6 | Epoched FutureFill | $9.30 \pm 0.02$ | $18.54 \pm 0.04$ | $36.84 \pm 0.04$ | $75.21 \pm 0.23$ | $182.49 \pm 0.31$ | $521.45 \pm 0.04$ |
| 515.46M | 1024 | 8 | Epoched FutureFill | $11.73$ | $23.31 \pm 0.02$ | $46.61 \pm 0.09$ | $96.42 \pm 0.38$ | $237.20$ | $684.54 \pm 0.10$ |
| 670.75M | 1024 | 12 | Epoched FutureFill | $17.07 \pm 0.01$ | $34.07$ | $68.07 \pm 0.15$ | $140.32 \pm 0.16$ | $347.01 \pm 0.09$ | $1009.36 \pm 0.02$ |
| 160.71M | 512 | 6 | Baseline | $7.49 \pm 0.02$ | $15.02 \pm 0.05$ | $30.24 \pm 0.07$ | $68.79 \pm 0.11$ | $187.53 \pm 0.10$ | $530.74 \pm 0.23$ |
| 180.13M | 512 | 8 | Baseline | $9.61 \pm 0.01$ | $19.33 \pm 0.01$ | $38.82 \pm 0.05$ | $88.87 \pm 0.18$ | $244.50 \pm 0.19$ | $698.11 \pm 0.08$ |
| 218.98M | 512 | 12 | Baseline | $13.85 \pm 0.01$ | $27.83 \pm 0.03$ | $55.960 \pm 0.07$ | $129.75 \pm 0.05$ | $360.35 \pm 0.17$ | $1035.73 \pm 0.24$ |
| 357.62M | 896 | 6 | Baseline | $7.87 \pm 0.01$ | $15.96 \pm 0.01$ | $35.68 \pm 0.03$ | $92.39 \pm 0.07$ | $265.51 \pm 0.09$ | $796.94 \pm 0.06$ |
| 417.08M | 896 | 8 | Baseline | $10.20 \pm 0.01$ | $20.61 \pm 0.01$ | $46.13 \pm 0.04$ | $119.85 \pm 0.06$ | $346.97 \pm 0.03$ | $1048.48 \pm 0.06$ |
| 535.99M | 896 | 12 | Baseline | $14.51 \pm 0.01$ | $29.48 \pm 0.03$ | $66.58 \pm 0.19$ | $174.97 \pm 0.02$ | $511.13 \pm 0.08$ | $1555.67 \pm 0.07$ |
| 437.81M | 1024 | 6 | Baseline | $8.00 \pm 0.01$ | $16.51$ | $38.19 \pm 0.02$ | $100.51$ | $292.20 \pm 0.03$ | $887.85 \pm 0.12$ |
| 515.46M | 1024 | 8 | Baseline | $10.33 \pm 0.01$ | $21.18 \pm 0.02$ | $49.05 \pm 0.06$ | $130.20 \pm 0.04$ | $381.39 \pm 0.03$ | $1167.59 \pm 0.11$ |
| 670.75M | 1024 | 12 | Baseline | $14.72 \pm 0.02$ | $30.33 \pm 0.05$ | $70.91 \pm 0.08$ | $190.12 \pm 0.12$ | $561.88 \pm 0.04$ | $1731.67 \pm 0.1$ |

Table 3: Decode time (in s) for STU-only models, without prefill.

| Parameter count | Input dim | Layer count | Cache Type | Decode time at generation length $L_{gen}$ | | | | | |
|---|---|---|---|---|---|---|---|---|---|
| | | | | 4096 | 8192 | 16384 | 32768 | 65536 | 126976 |
| 163.03M | 512 | 6 | Epoched FutureFill | 9.51 | $18.91 \pm 0.10$ | $38.00 \pm 0.02$ | $75.45 \pm 0.03$ | $151.69 \pm 0.40$ | $293.22 \pm 0.04$ |
| 183.23M | 512 | 8 | Epoched FutureFill | $12.11 \pm 0.01$ | $24.01 \pm 0.02$ | $48.41 \pm 0.05$ | $96.58 \pm 0.31$ | $193.16 \pm 0.01$ | $370.67 \pm 0.24$ |
| 223.63M | 512 | 12 | Epoched FutureFill | $16.89 \pm 0.02$ | $33.73 \pm 0.01$ | $67.22 \pm 0.11$ | $134.71 \pm 0.16$ | $268.93 \pm 0.08$ | $526.35 \pm 0.64$ |
| 364.78M | 896 | 6 | Epoched FutureFill | $9.83 \pm 0.03$ | 19.68 | $39.30 \pm 0.02$ | $78.50 \pm 0.01$ | $156.52 \pm 0.45$ | $335.21 \pm 1.83$ |
| 426.63M | 896 | 8 | Epoched FutureFill | $12.25 \pm 0.02$ | $24.53 \pm 0.05$ | $48.93 \pm 0.03$ | $98.08 \pm 0.07$ | $194.62 \pm 0.03$ | $430.45 \pm 0.02$ |
| 550.31M | 896 | 12 | Epoched FutureFill | $17.46 \pm 0.02$ | $34.85 \pm 0.01$ | $69.58 \pm 0.34$ | $138.96 \pm 0.20$ | $278.09 \pm 0.01$ | $627.05 \pm 0.09$ |
| 447.17M | 1 024 | 6 | Epoched FutureFill | $9.64 \pm 0.01$ | $19.24 \pm 0.08$ | 38.53 | $76.79 \pm 0.27$ | $152.75 \pm 0.11$ | $350.96 \pm 0.05$ |
| 527.94M | 1 024 | 8 | Epoched FutureFill | $12.21 \pm 0.01$ | $24.44 \pm 0.01$ | 48.74 | $97.09 \pm 0.09$ | $196.06 \pm 0.49$ | $457.30 \pm 0.23$ |
| 689.48M | 1 024 | 12 | Epoched FutureFill | 17.50 | $34.93 \pm 0.01$ | $69.70 \pm 0.13$ | $139.12 \pm 0.72$ | $281.83 \pm 0.09$ | $668.60 \pm 0.16$ |
| 163.03M | 512 | 6 | Baseline (naïve conv) | $8.85 \pm 0.01$ | $17.64 \pm 0.10$ | $35.49 \pm 0.08$ | $70.49 \pm 0.04$ | $145.73 \pm 0.22$ | $337.90 \pm 0.09$ |
| 183.23M | 512 | 8 | Baseline | $10.97 \pm 0.01$ | $21.96 \pm 0.03$ | $43.85 \pm 0.11$ | $87.47 \pm 0.09$ | $183.71 \pm 0.06$ | $438.86 \pm 0.10$ |
| 223.63M | 512 | 12 | Baseline | $15.45 \pm 0.02$ | $31.00 \pm 0.11$ | $61.99 \pm 0.16$ | $123.79 \pm 0.03$ | $264.36 \pm 0.16$ | $642.55 \pm 0.30$ |
| 364.78M | 896 | 6 | Baseline | $8.95 \pm 0.01$ | $17.89 \pm 0.01$ | $35.81 \pm 0.17$ | $72.60 \pm 0.02$ | $171.12 \pm 0.14$ | $457.10 \pm 0.40$ |
| 426.63M | 896 | 8 | Baseline | $11.24 \pm 0.01$ | $22.54 \pm 0.07$ | $44.99 \pm 0.03$ | $92.92 \pm 0.05$ | $223.65 \pm 0.04$ | $601.28 \pm 0.28$ |
| 550.31M | 896 | 12 | Baseline | $15.94 \pm 0.01$ | $31.91 \pm 0.02$ | $63.39 \pm 0.20$ | $132.32 \pm 0.22$ | $324.83 \pm 0.04$ | $885.40 \pm 0.15$ |
| 447.17M | 1 024 | 6 | Baseline | $8.88 \pm 0.02$ | $17.75 \pm 0.03$ | $35.40 \pm 0.16$ | $73.99 \pm 0.06$ | $182.20 \pm 0.19$ | $498.67 \pm 0.32$ |
| 527.94M | 1 024 | 8 | Baseline | $11.27 \pm 0.02$ | $22.55 \pm 0.02$ | $45.12 \pm 0.03$ | $95.20 \pm 0.02$ | $237.16 \pm 0.17$ | $654.97 \pm 0.01$ |
| 689.48M | 1 024 | 12 | Baseline | $15.96 \pm 0.02$ | $31.90 \pm 0.01$ | $63.71 \pm 0.12$ | $136.66 \pm 0.14$ | $346.37 \pm 0.06$ | $967.64 \pm 0.12$ |

Table 4: Decode time (in s) for Hybrid models (50% STU / 50% Attention), without prefill.

## B.3 ADDITIONAL ABLATIONS, WITH PREFILL

Prefill times reported below have been measured separately from generation times, i.e. generation times below do not include prefill times.

### B.3.1 PREFILL LENGTH IS LARGER THAN (OR EQUAL TO) GENERATION LENGTH

During prefill, the minimal FFT length required to recover the linear convolution of a prompt of length $L_{\text{prompt}}$ and a generation of length $L_{\text{generation}}$ is

$$N = \text{next\_pow2}\big(L_{\text{prompt}} + \big(L_{\text{prompt}} + L_{\text{generation}}\big) - 1\big).$$

For $L_{\text{prompt}} = 16\,384$ and $L_{\text{generation}} \in \{4\,096, \dots, 32\,768\}$, this yields $N = 65\,536$. This is why Table 5 reports only the average prefill time and its sample standard deviation. Likewise, for $L_{\text{prompt}} = 32\,768$ and $L_{\text{generation}} \in \{4\,096, \dots, 16\,384\}$, the minimal FFT size is $N = 131\,072$, and Table 6 summarizes the corresponding average prefill times and their sample standard deviations.

| Parameter count | Input dim | Layer count | Cache Type | Prefill Time | Decode time at generation length $L_{gen}$ | | | |
|---|---|---|---|---|---|---|---|---|
| | | | | | 4096 | 8192 | 16384 | 32768 |
| 160.71M | 512 | 6 | Epoched FutureFill | 4.03 | $10.18 \pm 0.01$ | $20.34 \pm 0.03$ | $40.39 \pm 0.01$ | $80.21 \pm 0.07$ |
| 180.13M | 512 | 8 | Epoched FutureFill | 5.35 | 13.08 | 26.19 | $51.43 \pm 0.01$ | $103.80 \pm 0.21$ |
| 218.98M | 512 | 12 | Epoched FutureFill | $8.01 \pm 0.02$ | $18.90 \pm 0.01$ | $37.46 \pm 0.02$ | $75.07 \pm 0.08$ | $149.97 \pm 0.08$ |
| 357.62M | 896 | 6 | Epoched FutureFill | $7.04 \pm 0.01$ | $10.10 \pm 0.01$ | $20.18 \pm 0.04$ | $40.46 \pm 0.24$ | $80.75 \pm 0.11$ |
| 417.08M | 896 | 8 | Epoched FutureFill | 9.36 | $13.01 \pm 0.02$ | $26.04 \pm 0.05$ | $52.35 \pm 0.09$ | $104.38 \pm 0.13$ |
| 437.81M | 1024 | 6 | Epoched FutureFill | $8.09 \pm 0.02$ | $10.10 \pm 0.03$ | $20.27 \pm 0.02$ | 40.36 | $80.65 \pm 0.06$ |
| 515.46M | 1024 | 8 | Epoched FutureFill | $10.74 \pm 0.03$ | $13.08 \pm 0.01$ | $26.22 \pm 0.04$ | $52.37 \pm 0.02$ | $104.91 \pm 0.09$ |
| 535.99M | 896 | 12 | Epoched FutureFill | $13.96 \pm 0.02$ | 19.03 | $37.64 \pm 0.16$ | $76.09 \pm 0.28$ | $152.01 \pm 0.06$ |
| 670.75M | 1024 | 12 | Epoched FutureFill | $16.01 \pm 0.02$ | $18.91 \pm 0.03$ | $37.71 \pm 0.16$ | $75.99 \pm 0.07$ | $152.27 \pm 0.20$ |
| 160.71M | 512 | 6 | Baseline | $4.03 \pm 0.01$ | 8.52 | 17.82 | $39.09 \pm 0.01$ | 92.37 |
| 180.13M | 512 | 8 | Baseline | 5.36 | $10.99 \pm 0.02$ | $23.13 \pm 0.01$ | 50.94 | $120.74 \pm 0.05$ |
| 218.98M | 512 | 12 | Baseline | 8.02 | $16.17 \pm 0.01$ | 33.96 | $74.95 \pm 0.04$ | $178.56 \pm 0.01$ |
| 357.62M | 896 | 6 | Baseline | 7.056 | 12.17 | 25.78 | $57.09 \pm 0.02$ | $134.20 \pm 0.03$ |
| 417.08M | 896 | 8 | Baseline | 9.37 | $15.84 \pm 0.01$ | 33.73 | 74.74 | $175.99 \pm 0.05$ |
| 437.81M | 1024 | 6 | Baseline | 8.09 | $13.44 \pm 0.02$ | $28.40 \pm 0.01$ | $62.73 \pm 0.02$ | $147.74 \pm 0.05$ |
| 515.46M | 1024 | 8 | Baseline | 10.74 | $17.49 \pm 0.01$ | $37.03 \pm 0.03$ | $81.94 \pm 0.02$ | $193.78 \pm 0.01$ |
| 535.99M | 896 | 12 | Baseline | 14.00 | $23.19 \pm 0.01$ | 49.34 | $109.55 \pm 0.04$ | $259.24 \pm 0.02$ |
| 670.75M | 1024 | 12 | Baseline | 16.05 | 25.63 | $54.37 \pm 0.03$ | $120.34 \pm 0.04$ | $285.88 \pm 0.13$ |

Table 5: Decode time (in s) for STU-only models, with prefill on an input prompt of length $L_{\text{prompt}} = 16{,}384$ tokens.

| Parameter count | Input dim | Layer count | Cache Type | Prefill Time | Decode time at generation length $L_{gen}$ | | |
|---|---|---|---|---|---|---|---|
| | | | | | 4096 | 8192 | 16384 |
| 160.71M | 512 | 6 | Epoched FutureFill | 8.01 | 10.11 ± 0.01 | 20.10 ± 0.02 | 40.02 ± 0.08 |
| 180.13M | 512 | 8 | Epoched FutureFill | 10.58 ± 0.01 | 13.07 ± 0.01 | 25.96 | 52.11 ± 0.19 |
| 218.98M | 512 | 12 | Epoched FutureFill | 15.81 ± 0.03 | 18.93 ± 0.01 | 37.70 ± 0.04 | 75.89 ± 0.02 |
| 357.62M | 896 | 6 | Epoched FutureFill | 14.01 ± 0.05 | 10.05 ± 0.01 | 20.13 | 40.26 ± 0.04 |
| 417.08M | 896 | 8 | Epoched FutureFill | 18.66 ± 0.04 | 13.18 ± 0.01 | 26.29 ± 0.04 | 52.36 ± 0.04 |
| 437.81M | 1024 | 6 | Epoched FutureFill | 16.05 ± 0.03 | 10.11 | 20.15 | 40.33 ± 0.05 |
| 515.46M | 1024 | 8 | Epoched FutureFill | 21.40 ± 0.03 | 13.12 ± 0.05 | 26.18 ± 0.02 | 52.22 ± 0.09 |
| 535.99M | 896 | 12 | Epoched FutureFill | 27.74 | 18.93 ± 0.36 | 37.89 ± 0.04 | 75.90 ± 0.07 |
| 670.75M | 1024 | 12 | Epoched FutureFill | 31.98 ± 0.09 | 19.06 ± 0.10 | 37.80 ± 0.01 | 75.66 ± 0.61 |
| 160.71M | 512 | 6 | Baseline | 7.99 | 12.08 | 24.93 ± 0.01 | 53.43 ± 0.01 |
| 180.13M | 512 | 8 | Baseline | 10.60 | 15.85 | 32.83 ± 0.02 | 69.82 |
| 218.98M | 512 | 12 | Baseline | 15.84 | 23.39 ± 0.01 | 48.50 ± 0.01 | 103.64 ± 0.02 |
| 357.62M | 896 | 6 | Baseline | 14.03 | 17.48 | 36.20 ± 0.01 | 77.18 ± 0.01 |
| 417.08M | 896 | 8 | Baseline | 18.62 | 22.96 | 47.51 | 101.50 ± 0.02 |
| 437.81M | 1024 | 6 | Baseline | 16.05 | 19.22 | 39.83 | 85.10 ± 0.01 |
| 515.46M | 1024 | 8 | Baseline | 21.28 | 25.23 | 52.31 ± 0.02 | 111.92 ± 0.06 |
| 535.99M | 896 | 12 | Baseline | 27.82 | 33.83 ± 0.01 | 70.03 ± 0.01 | 149.86 ± 0.02 |
| 670.75M | 1024 | 12 | Baseline | 32.70 | 37.21 ± 0.02 | 77.15 ± 0.02 | 165.13 ± 0.07 |

Table 6: Decode time (in s) for STU-only models, with prefill on an input prompt of length $L_{\text{prompt}} = 32{,}768$ tokens.

### B.3.2 PREFILL LENGTH IS SMALLER THAN (OR EQUAL TO) GENERATION LENGTH

In the ablations shown in this section, the prompt length is fixed at $L_{\text{prompt}}$, the column headers refer to the total sequence length $L = L_{prompt} + L_{generation}$, rather than to the generation length $L_{\text{gen}}$ alone. For instance, in the case of $L_{\text{prompt}} = 512$: $L = 8192$ means that there are 512 tokens in the input prompt and 7680 generated tokens.

| Parameter count | Input dim | Layer count | Cache Type | Total length $L(= L_{gen} + L_{prompt})$ | | | | | |
|---|---|---|---|---|---|---|---|---|---|
| | | | | 4096 | 8192 | 16384 | 32768 | 65536 | 131072 |
| 180.13M | 512 | 8 | Epoched FutureFill | 11.45 ± 0.03 | 24.60 ± 0.02 | 50.71 ± 0.17 | 103.12 ± 0.04 | 206.14 ± 1.86 | 461.63 ± 0.49 |
| 218.98M | 512 | 12 | Epoched FutureFill | 16.16 ± 0.01 | 34.49 ± 0.01 | 71.64 ± 0.08 | 145.42 ± 0.60 | 293.34 ± 0.35 | 671.33 ± 0.61 |
| 417.08M | 896 | 8 | Epoched FutureFill | 11.24 ± 0.03 | 23.91 ± 0.08 | 49.54 ± 0.07 | 100.51 ± 0.40 | 228.71 ± 0.12 | 659.71 ± 0.21 |
| 535.99M | 896 | 12 | Epoched FutureFill | 16.22 ± 0.07 | 34.49 ± 0.08 | 71.42 ± 0.01 | 145.93 ± 0.09 | 334.20 ± 0.17 | 976.03 ± 1.00 |
| 515.46M | 1024 | 8 | Epoched FutureFill | 11.13 ± 0.01 | 23.92 ± 0.10 | 49.57 ± 0.02 | 101.80 ± 0.27 | 245.07 ± 0.45 | 735.22 ± 0.11 |
| 670.75M | 1024 | 12 | Epoched FutureFill | 16.33 ± 0.02 | 35.10 ± 0.03 | 72.42 ± 0.10 | 149.12 ± 0.07 | 361.35 ± 0.29 | 1097.50 |
| 180.13M | 512 | 8 | Baseline | 8.50 ± 0.02 | 18.26 ± 0.01 | 37.61 ± 0.09 | 88.27 ± 0.12 | 245.27 ± 0.01 | 738.23 ± 0.92 |
| 218.98M | 512 | 12 | Baseline | 11.90 ± 0.01 | 25.60 | 53.23 ± 0.04 | 127.73 ± 0.14 | 359.73 ± 0.25 | 1094.02 ± 0.01 |
| 417.08M | 896 | 8 | Baseline | 8.83 ± 0.01 | 19.20 ± 0.04 | 44.95 ± 0.02 | 119.35 ± 0.08 | 347.60 ± 0.09 | 1111.11 ± 0.08 |
| 535.99M | 896 | 12 | Baseline | 12.59 ± 0.01 | 27.38 | 64.91 ± 0.06 | 174.19 ± 0.08 | 512.10 ± 0.05 | 1648.08 ± 0.21 |
| 515.46M | 1024 | 8 | Baseline | 8.94 | 19.87 ± 0.05 | 48.11 ± 0.03 | 129.78 ± 0.02 | 382.44 ± 0.04 | 1238.22.86 ± 0.02 |
| 670.75M | 1024 | 12 | Baseline | 12.68 ± 0.01 | 28.42 ± 0.02 | 69.42 ± 0.13 | 189.57 ± 0.10 | 563.47 ± 0.14 | 1837.16 ± 0.05 |

Table 7: Decode time (in s) for STU-only models, with prefill on an input prompt of length $L_{prompt} = 512$ tokens.

| Parameter count | Input dim | Layer count | Cache Type | Prefill rimes associated with total length $L(= L_{gen} + L_{prompt})$ | | | | | |
|---|---|---|---|---|---|---|---|---|---|
| | | | | 4096 | 8192 | 16384 | 32768 | 65536 | 131072 |
| 180.13M | 512 | 8 | Epoched FutureFill | 0.20 | 0.20 | 0.21 | 0.21 | 0.25 | 0.38 |
| 218.98M | 512 | 12 | Epoched FutureFill | 0.30 | 0.30 | 0.31 | 0.32 | 0.36 | 0.57 |
| 417.08M | 896 | 8 | Epoched FutureFill | 0.39 | 0.39 | 0.40 | 0.41 | 0.46 | 0.70 |
| 535.99M | 896 | 12 | Epoched FutureFill | 0.58 | 0.58 | 0.59 | 0.62 | 0.69 | 1.05 |
| 515.46M | 1024 | 8 | Epoched FutureFill | 0.39 | 0.40 | 0.41 | 0.42 | 0.48 | 0.75 |
| 670.75M | 1024 | 12 | Epoched FutureFill | 0.59 | 0.59 | 0.60 | 0.63 | 0.72 | 1.13 |
| 180.13M | 512 | 8 | Baseline | 0.20 | 0.20 | 0.20 | 0.20 | 0.20 | 0.20 |
| 218.98M | 512 | 12 | Baseline | 0.30 | 0.30 | 0.30 | 0.30 | 0.30 | 0.30 |
| 417.08M | 896 | 8 | Baseline | 0.38 | 0.38 | 0.38 | 0.38 | 0.38 | 0.38 |
| 535.99M | 896 | 12 | Baseline | 0.57 | 0.57 | 0.57 | 0.57 | 0.57 | 0.57 |
| 515.46M | 1024 | 8 | Baseline | 0.39 | 0.39 | 0.39 | 0.39 | 0.39 | 0.39 |
| 670.75M | 1024 | 12 | Baseline | 0.58 | 0.58 | 0.58 | 0.58 | 0.58 | 0.58 |

Table 8: Prefill time (in s) for STU-only models, associated with an input prompt of length $L_{prompt} = 512$ tokens.

| Parameter count | Input dim | Layer count | Cache Type | Total length $L (= L_{gen} + L_{prompt})$ | | | | | |
|---|---|---|---|---|---|---|---|---|---|
| | | | | 4096 | 8192 | 16384 | 32768 | 65536 | 131072 |
| 180.13M | 512 | 8 | Epoched FutureFill | 9.77 ± 0.05 | 22.73 ± 0.11 | 48.54 ± 0.20 | 101.12 ± 0.35 | 205.98 ± 1.05 | 460.10 ± 1.05 |
| 218.98M | 512 | 12 | Epoched FutureFill | 13.86 ± 0.02 | 32.30 ± 0.06 | 69.19 ± 0.04 | 143.38 ± 0.03 | 291.17 ± 0.62 | 670.61 ± 1.28 |
| 417.08M | 896 | 8 | Epoched FutureFill | 9.68 ± 0.02 | 22.32 ± 0.02 | 48.05 ± 0.03 | 99.31 ± 0.21 | 226.30 ± 0.20 | 658.32 ± 0.30 |
| 535.99M | 896 | 12 | Epoched FutureFill | 13.92 ± 0.04 | 32.32 ± 0.09 | 69.49 ± 0.01 | 144.03 ± 0.11 | 330.52 ± 0.61 | 974.19 ± 0.54 |
| 515.46M | 1024 | 8 | Epoched FutureFill | 9.57 ± 0.03 | 22.43 ± 0.03 | 47.71 ± 0.13 | 100.29 ± 0.30 | 242.70 ± 0.20 | 733.30 ± 0.12 |
| 670.75M | 1024 | 12 | Epoched FutureFill | 13.63 ± 0.02 | 31.80 ± 0.07 | 68.53 ± 0.04 | 143.44 ± 0.49 | 355.29 ± 0.15 | 1099.61 ± 2.16 |
| 180.13M | 512 | 8 | Baseline | 7.13 ± 0.01 | 16.62 ± 0.03 | 35.82 ± 0.07 | 86.23 ± 0.09 | 242.88 ± 0.12 | 736.30 ± 0.13 |
| 218.98M | 512 | 12 | Baseline | 10.27 ± 0.01 | 23.91 ± 0.01 | 51.78 ± 0.12 | 126.30 ± 0.03 | 357.94 ± 0.41 | 1091.89 ± 0.35 |
| 417.08M | 896 | 8 | Baseline | 7.60 ± 0.01 | 17.94 ± 0.03 | 43.71 ± 0.06 | 118.07 ± 0.08 | 46.26 ± 0.04 | 1109.39 ± 0.09 |
| 535.99M | 896 | 12 | Baseline | 10.83 ± 0.01 | 25.61 ± 0.02 | 63.14 ± 0.13 | 172.44 ± 0.03 | 509.91 ± 0.24 | 1645.93 ± 0.11 |
| 515.46M | 1024 | 8 | Baseline | 7.68 | 18.64 | 46.86 ± 0.05 | 128.61 ± 0.12 | 381.15 ± 0.38 | 1237.62 ± 0.11 |
| 670.75M | 1024 | 12 | Baseline | 10.90 | 26.58 ± 0.04 | 67.62 ± 0.01 | 187.76 ± 0.07 | 561.71 ± 0.01 | 1834.86 ± 0.18 |

Table 9: Decode time (in s) for STU-only models, with prefill on an input prompt of length $L_{prompt} = 1024$ tokens.

| Parameter count | Input dim | Layer count | Cache Type | Prefill times associated with total length $L (= L_{gen} + L_{prompt})$ | | | | | |
|---|---|---|---|---|---|---|---|---|---|
| | | | | 4096 | 8192 | 16384 | 32768 | 65536 | 131072 |
| 180.13M | 512 | 8 | Epoched FutureFill | 0.38 | 0.39 | 0.39 ± 0.09 | 0.40 | 0.43 | 0.56 |
| 218.98M | 512 | 12 | Epoched FutureFill | 0.58 | 0.58 | 0.59 | 0.60 | 0.64 | 0.85 |
| 417.08M | 896 | 8 | Epoched FutureFill | 0.66 | 0.67 | 0.68 | 0.69 | 0.74 | 0.98 |
| 535.99M | 896 | 12 | Epoched FutureFill | 0.99 | 1.00 | 1.01 | 1.03 | 1.10 | 1.47 |
| 515.46M | 1024 | 8 | Epoched FutureFill | 0.76 | 0.76 | 0.77 | 0.79 | 0.84 | 1.12 |
| 670.75M | 1024 | 12 | Epoched FutureFill | 1.13 | 1.14 | 1.15 | 1.18 | 1.26 | 1.68 |
| 180.13M | 512 | 8 | Baseline | 0.38 | 0.38 | 0.38 | 0.38 | 0.38 | 0.38 |
| 218.98M | 512 | 12 | Baseline | 0.58 | 0.58 | 0.58 | 0.58 | 0.58 | 0.58 |
| 417.08M | 896 | 8 | Baseline | 0.66 | 0.66 | 0.66 | 0.66 | 0.66 | 0.66 |
| 535.99M | 896 | 12 | Baseline | 0.99 | 0.99 | 0.99 | 0.99 | 0.99 | 0.99 |
| 515.46M | 1024 | 8 | Baseline | 0.75 | 0.75 | 0.75 | 0.75 | 0.75 | 0.75 |
| 670.75M | 1024 | 12 | Baseline | 1.12 | 1.12 | 1.12 | 1.12 | 1.12 | 1.12 |

Table 10: Prefill time (in s) for STU-only models, associated with an input prompt of length $L_{prompt} = 1024$ tokens.

| Parameter count | Input dim | Layer count | Cache Type | Total length $L (= L_{gen} + L_{prompt})$ | | | | | |
|---|---|---|---|---|---|---|---|---|---|
| | | | | 4096 | 8192 | 16384 | 32768 | 65536 | 131072 |
| 180.13M | 512 | 8 | Epoched FutureFill | 6.44 ± 0.01 | 19.32 ± 0.03 | 44.96 ± 0.11 | 95.46 ± 0.09 | 195.44 ± 0.71 | 450.41 ± 0.70 |
| 218.98M | 512 | 12 | Epoched FutureFill | 9.23 ± 0.02 | 27.81 ± 0.05 | 64.44 ± 0.13 | 137.98 ± 0.56 | 286.83 ± 0.11 | 662.80 ± 0.35 |
| 417.08M | 896 | 8 | Epoched FutureFill | 6.39 ± 0.03 | 19.22 ± 0.06 | 44.82 ± 0.06 | 95.73 ± 0.52 | 222.07 ± 0.51 | 649.40 ± 0.12 |
| 535.99M | 896 | 12 | Epoched FutureFill | 9.27 ± 0.02 | 27.78 ± 0.12 | 64.48 ± 0.32 | 138.90 ± 0.46 | 323.72 ± 0.03 | 955.18 ± 0.43 |
| 515.46M | 1024 | 8 | Epoched FutureFill | 6.38 ± 0.03 | 19.11 ± 0.03 | 44.82 ± 0.12 | 96.74 ± 0.10 | 237.19 ± 0.20 | 722.81 ± 0.32 |
| 670.75M | 1024 | 12 | Epoched FutureFill | 9.09 ± 0.01 | 27.20 ± 0.08 | 63.28 ± 0.30 | 138.06 ± 0.72 | 346.51 ± 0.26 | 1077.25 ± 0.62 |
| 180.13M | 512 | 8 | Baseline | 4.75 | 14.27 | 33.51 ± 0.07 | 84.11 ± 0.08 | 240.56 ± 0.11 | 733.73 ± 0.05 |
| 218.98M | 512 | 12 | Baseline | 6.84 | 20.56 ± 0.01 | 48.35 ± 0.06 | 122.67 ± 0.13 | 354.58 ± 0.11 | 1088.95 ± 0.13 |
| 417.08M | 896 | 8 | Baseline | 5.06 | 15.42 ± 0.01 | 41.17 ± 0.04 | 115.49 ± 0.01 | 343.68 | 1107.32 ± 0.03 |
| 535.99M | 896 | 12 | Baseline | 7.22 ± 0.01 | 22.08 ± 0.03 | 59.62 ± 0.04 | 168.82 ± 0.08 | 506.61 ± 0.15 | 1642.43 ± 0.11 |
| 515.46M | 1024 | 8 | Baseline | 5.13 ± 0.01 | 16.09 ± 0.01 | 44.29 ± 0.02 | 126.11 ± 0.02 | 378.62 ± 0.05 | 1234.521 ± 0.04 |
| 670.75M | 1024 | 12 | Baseline | 7.23 ± 0.02 | 22.94 ± 0.02 | 64.01 ± 0.05 | 184.11 ± 0.04 | 557.94 ± 0.41 | 1831.83 ± 0.38 |

Table 11: Decode time (in s) for STU-only models, with prefill on an input prompt of length $L_{prompt} = 2048$ tokens.

| Parameter count | Input dim | Layer count | Cache Type | Prefill times associated with total length $L (= L_{gen} + L_{prompt})$ | | | | | |
|---|---|---|---|---|---|---|---|---|---|
| | | | | 4096 | 8192 | 16384 | 32768 | 65536 | 131072 |
| 180.13M | 512 | 8 | Epoched FutureFill | 0.72 | 0.73 | 0.74 | 0.74 | 0.77 | 0.91 |
| 218.98M | 512 | 12 | Epoched FutureFill | 1.09 | 1.09 | 1.10 | 1.11 | 1.15 | 1.36 |
| 417.08M | 896 | 8 | Epoched FutureFill | 1.27 | 1.27 | 1.28 | 1.29 | 1.34 | 1.58 |
| 535.99M | 896 | 12 | Epoched FutureFill | 1.89 | 1.90 | 1.91 | 1.93 | 2.00 | 2.36 |
| 515.46M | 1024 | 8 | Epoched FutureFill | 1.44 | 1.45 | 1.46 | 1.48 | 1.53 | 1.81 |
| 670.75M | 1024 | 12 | Epoched FutureFill | 2.16 | 2.17 | 2.18 | 2.21 | 2.29 | 2.70 |
| 180.13M | 512 | 8 | Baseline | 0.73 | 0.73 | 0.73 | 0.73 | 0.73 | 0.73 |
| 218.98M | 512 | 12 | Baseline | 1.09 | 1.09 | 1.09 | 1.09 | 1.09 | 1.09 |
| 417.08M | 896 | 8 | Baseline | 1.26 | 1.26 | 1.26 | 1.26 | 1.26 | 1.26 |
| 535.99M | 896 | 12 | Baseline | 1.88 | 1.88 | 1.88 | 1.88 | 1.88 | 1.88 |
| 515.46M | 1024 | 8 | Baseline | 1.44 | 1.44 | 1.44 | 1.44 | 1.44 | 1.44 |
| 670.75M | 1024 | 12 | Baseline | 2.14 | 2.14 | 2.14 | 2.14 | 2.14 | 2.14 |

Table 12: Prefill time (in s) for STU-only models, associated with an input prompt of length $L_{prompt} = 2048$ tokens.

| Parameter count | Input dim | Layer count | Cache Type | Total length $L(= L_{gen} + L_{prompt})$ | | | | |
|---|---|---|---|---|---|---|---|---|
| | | | | 8192 | 16384 | 32768 | 65536 | 131072 |
| 180.13M | 512 | 8 | Epoched FutureFill | $12.70 \pm 0.01$ | $37.98 \pm 0.11$ | $88.26 \pm 0.12$ | $189.96 \pm 0.51$ | $438.92 \pm 0.69$ |
| 218.98M | 512 | 12 | Epoched FutureFill | 18.58 | $55.46 \pm 0.03$ | $129.42 \pm 0.15$ | $277.51 \pm 0.29$ | $645.49 \pm 0.29$ |
| 417.08M | 896 | 8 | Epoched FutureFill | $12.81 \pm 0.01$ | $38.27 \pm 0.19$ | $89.61 \pm 0.13$ | $212.41 \pm 0.16$ | $630.59 \pm 0.20$ |
| 535.99M | 896 | 12 | Epoched FutureFill | 18.53 | $55.65 \pm 0.01$ | $130.20 \pm 0.30$ | $307.42 \pm 0.13$ | $923.36 \pm 0.33$ |
| 515.46M | 1024 | 8 | Epoched FutureFill | $12.77 \pm 0.03$ | $38.28 \pm 0.22$ | $89.17 \pm 0.77$ | $226.41 \pm 0.02$ | $701.11 \pm 0.12$ |
| 670.75M | 1024 | 12 | Epoched FutureFill | $18.67 \pm 0.01$ | $56.04 \pm 0.12$ | $129.21 \pm 0.52$ | $326.18 \pm 0.05$ | $1028.72 \pm 1.43$ |
| 180.13M | 512 | 8 | Baseline | $9.50 \pm 0.02$ | $28.67 \pm 0.15$ | 79.24 | $235.57 \pm 0.12$ | $729.12 \pm 0.07$ |
| 218.98M | 512 | 12 | Baseline | 13.69 | $41.50 \pm 0.04$ | $116.01 \pm 0.04$ | $347.99 \pm 0.25$ | $1082.01 \pm 0.09$ |
| 417.08M | 896 | 8 | Baseline | 10.37 | $36.16 \pm 0.02$ | $110.52 \pm 0.02$ | $338.62 \pm 0.04$ | $1101.93 \pm 0.05$ |
| 535.99M | 896 | 12 | Baseline | $14.82 \pm 0.01$ | 52.33 | $161.64 \pm 0.04$ | $499.29 \pm 0.13$ | $1635.09 \pm 0.10$ |
| 515.46M | 1024 | 8 | Baseline | $10.93 \pm 0.01$ | $39.15 \pm 0.03$ | $120.89 \pm 0.01$ | $373.59 \pm 0.07$ | $1229.29 \pm 0.15$ |
| 670.75M | 1024 | 12 | Baseline | 15.70 | $56.81 \pm 0.02$ | $176.90 \pm 0.15$ | $550.78 \pm 0.03$ | $1824.47 \pm 0.01$ |

Table 13: Decode time (in s) for STU-only models, with prefill on an input prompt of length $L_{prompt} = 4096$ tokens.

| Parameter count | Input dim | Layer count | Cache Type | Prefill times associated with total length $L(= L_{gen} + L_{prompt})$ | | | | |
|---|---|---|---|---|---|---|---|---|
| | | | | 8192 | 16384 | 32768 | 65536 | 131072 |
| 180.13M | 512 | 8 | Epoched FutureFill | 1.41 | 1.41 | 1.42 | 1.45 | 1.57 |
| 218.98M | 512 | 12 | Epoched FutureFill | 2.10 | 2.11 | 2.13 | 2.17 | 2.35 |
| 417.08M | 896 | 8 | Epoched FutureFill | 2.46 | 2.47 | 2.48 | 2.53 | 2.74 |
| 535.99M | 896 | 12 | Epoched FutureFill | 3.67 | 3.68 | 3.71 | 3.78 | 4.10 |
| 515.46M | 1024 | 8 | Epoched FutureFill | 2.81 | 2.81 | 2.83 | 2.89 | 3.13 |
| 670.75M | 1024 | 12 | Epoched FutureFill | 4.20 | 4.21 | 4.24 | 4.32 | 4.68 |
| 180.13M | 512 | 8 | Baseline | 1.40 | 1.40 | 1.40 | 1.40 | 1.40 |
| 218.98M | 512 | 12 | Baseline | 2.08 | 2.08 | 2.08 | 2.08 | 2.08 |
| 417.08M | 896 | 8 | Baseline | 2.45 | 2.45 | 2.45 | 2.45 | 2.45 |
| 535.99M | 896 | 12 | Baseline | 3.65 | 3.65 | 3.65 | 3.65 | 3.65 |
| 515.46M | 1024 | 8 | Baseline | 2.79 | 2.79 | 2.79 | 2.79 | 2.79 |
| 670.75M | 1024 | 12 | Baseline | 4.16 | 4.16 | 4.16 | 4.16 | 4.16 |

Table 14: Prefill time (in s) for STU-only models, associated with an input prompt of length $L_{prompt} = 4096$ tokens.

## B.4 Additional Ablations on the Epoched-FutureFill cache length $K$, without Prefill

| Parameter count | Input dim | Layer count | Cache Type | FutureFill cache length $K$ | | | | | |
|---|---|---|---|---|---|---|---|---|---|
| | | | | 128 | 256 | 512 | 1024 | 2048 | 4096 |
| 417.08M | 896 | 8 | Epoched FutureFill | $244.92 \pm 0.19$ | $232.66 \pm 0.08$ | $227.99 \pm 0.31$ | $\mathbf{226.28 \pm 0.18}$ | $228.75 \pm 0.04$ | $236.42 \pm 0.23$ |
| 535.99M | 896 | 12 | Epoched FutureFill | $357.80 \pm 0.16$ | $339.77 \pm 0.63$ | $332.89 \pm 0.50$ | $\mathbf{330.90 \pm 0.52}$ | $334.38 \pm 0.03$ | $345.00 \pm 0.10$ |
| 654.90M | 896 | 16 | Epoched FutureFill | $472.03 \pm 0.78$ | $447.05 \pm 0.05$ | $436.86 \pm 0.35$ | $\mathbf{434.49 \pm 0.40}$ | $439.29 \pm 0.46$ | $454.67 \pm 0.28$ |
| 515.465M | 1024 | 8 | Epoched FutureFill | $262.33 \pm 0.17$ | $248.88 \pm 0.57$ | $243.49 \pm 0.23$ | $\mathbf{241.69 \pm 0.10}$ | $244.61 \pm 0.45$ | $257.85 \pm 0.06$ |
| 670.75M | 1024 | 12 | Epoched FutureFill | $384.37 \pm 0.024$ | $364.04 \pm 0.52$ | $355.01 \pm 0.17$ | $\mathbf{352.96 \pm 0.20}$ | $358.316 \pm 0.15$ | $377.52 \pm 0.43$ |
| 826.05M | 1024 | 16 | Epoched FutureFill | $505.50 \pm 0.17$ | $478.47 \pm 0.2$ | $467.12 \pm 0.16$ | $\mathbf{463.74 \pm 0.35}$ | $471.32 \pm 0.14$ | $496.88 \pm 0.13$ |

Table 15: Decode time (in s) for STU-only models for a fixed generation length of $65{,}536$ tokens (without prefill).

## B.5 Additional Timing Breakdowns

| Generation length | 4096 | 8192 | 16384 | 16384 | 32768 | 65536 | 126976 |
|---|---|---|---|---|---|---|---|
| Prefill (in tokens) | 32768 | 32768 | 32768 | N/A | N/A | N/A | N/A |
| STU (= FFT) | 26.76 | 56.02 | 121.77 | 31.90 | 102.38 | 362.86 | 1299.10 |
| MLP | 5.80 | 11.78 | 24.22 | 22.04 | 49.47 | 111.93 | 242.95 |
| RMSNorm | 3.90 | 7.85 | 16.08 | 14.29 | 32.19 | 73.21 | 159.49 |
| Dropout | 0.63 | 1.27 | 2.61 | 2.28 | 5.19 | 11.80 | 25.80 |
| Embedding | 0.13 | 0.24 | 0.46 | 0.40 | 0.89 | 2.02 | 4.50 |

Table 16: Module-wise times (in s) vs. generation length (batch=1) for the 670.75M-parameter STU-only model. Baseline model (naive convolutions).

| Generation length | 4096 | 8192 | 16384 | 16384 | 32768 | 65536 | 126976 |
| :--- | :---: | :---: | :---: | :---: | :---: | :---: | :---: |
| **Prefill (in tokens)** | 32768 | 32768 | 32768 | N/A | N/A | N/A | N/A |
| STU (= FFT) | 9.79 | 19.41 | 38.83 | 35.26 | 73.06 | 191.76 | 637.01 |
| MLP | 5.16 | 10.28 | 20.69 | 18.21 | 37.42 | 86.89 | 209.14 |
| RMSNorm | 3.45 | 6.82 | 13.58 | 12.31 | 25.16 | 57.71 | 137.98 |
| Dropout | 0.55 | 1.10 | 2.19 | 1.95 | 3.97 | 9.06 | 21.50 |
| Embedding | 0.10 | 0.19 | 0.37 | 0.34 | 0.70 | 1.60 | 3.84 |

Table 17: Module-wise times (in s) vs. generation length (batch=1) for the 670.75M-parameter STU-only model, using FutureFill.

### B.6    Precision of the FFT & Downstream Tasks

Our current implementation uses PyTorch's built-in FFT, which currently supports only float32, float64, and float16. We perform ablations on a 330M parameter-pre-trained hybrid-STU model using end-to-end bfloat16 but upcasting the FFT to fp16/fp32 when needed, to show that it does not degrade accuracy. The baseline is kept end-to-end in bf16. The pretrained model is openly available at Hazan-Lab/FlashSTU-340M-0428.

| | WinoGrande | HellaSwag | PiQA |
| :--- | :---: | :---: | :---: |
| FutureFill, upcast to fp32 | 0.51381 | 0.3045 | 0.6553 |
| FutureFill, upcast to fp16 | 0.5090 | 0.3047 | 0.6551 |
| Baseline (convolution), bf16 | 0.5160 | 0.3046 | 0.6534 |

Table 18: Evaluation scores on downstream tasks for 330M hybrid-STU, using different precisions for the FFT. Other calculations and weights are kept in bf16.

## C    Experimental Comparison with Transformers

We experimentally evaluate Epoched-FutureFill (Algorithm 1) which has a runtime of $O(L^{3/2}\sqrt{\log L})$ and Continuous-FutureFill (Algorithm 2) which has a runtime of $O(L\log^2 L)$ against the naive implementation of convolution which has a runtime of $O(L^2)$ when generating $L$ tokens from scratch. We also provide a comparison with a self-attention based Transformer model (with a standard implementation of KV cache and with the same hidden dimension, number of layers and commensurately chosen other parameters, see next subsection for complete details on these models).

For increasing values of $L$, we measure the time it takes for the model to generate $L$ tokens from scratch (i.e. no prompt provided). In Figure 4 we plot the total generation time, as functions of $L$. We see the behavior that is expected: the naive decoder runs in total time $O(L^2)$, similar to the decoder for transformer while our method EpochedFutureFill is able to achieve a significant sub-quadratic improvement.

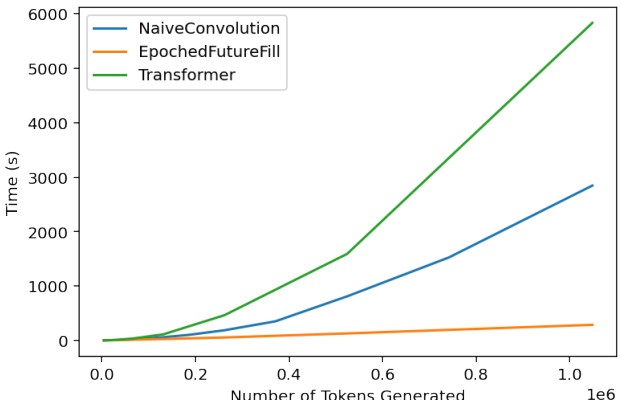

Figure 4: Total time for generating $L$ tokens, as a function of $L$.

In the next section we provide the details of our implementation.

## C.1 EXPERIMENT DETAILS

For our experiments we consider a two layer model with either multi-headed self-attention layers (referred to as Transformer) or STU layers (referred to as convolutional network). The hidden dimension $d$ (or the model dimension) of the networks are fixed to be 32, and for the Transformer we set the number of heads to be 4 and the key/value size to be 8. The networks do not have embedding or unembedding layers, and contain standard implementations of residual connections, layer-norms and a feed-forward (FFN) layer between every attention or STU layer. More information on the STU with tensordot approximation is available in Appendix B.1. The FFN layer used in the experiments is the FFN$_{\text{GeGLU}}$ layer proposed in Shazeer (2020). For the Transformer we employ a standard implementation of KV-cache for efficiency (i.e. caching the KV values of previously generated tokens for every attention layer).

Since we have equated the hidden dimensionality of the network across all our settings, we can see that, naively computed, the number of flops per token of both the Transformer as well as the convolutional model are of the same order which is also observed in the experiments.

Finally the experiments in this section are implemented in Jax (Bradbury et al., 2018) were performed on a single Google TPUv2 machine (Jouppi et al., 2020).

## D EXTENDED RELATED WORK AND DETAILS ON CONVOLUTIONAL SEQUENCE PREDICTION MODELS

### D.1 RELATED WORK

**State space models and convolutional sequence prediction.** Recurrent neural networks have been revisited in recent deep learning literature for sequence prediction in the form of state space models (SSMs), many of which can be parameterized as convolutional models. Gu et al. (2020) propose the HiPPO framework for continuous-time memorization, and shows that with a special class of system matrices $A$ (HiPPO matrices), SSMs have the capacity for long-range memory. Later works (Gu et al., 2021b;a; Gupta et al., 2022; Smith et al., 2023) focus on removing nonlinearities and devising computationally efficient methods that are also numerically stable. To improve the performance of SSMs on language modeling tasks Dao et al. (2022b) propose architectural changes as well as FFT algorithms with better hardware utilization, to close the speed gap between SSMs and Transformers. Further investigation in Orvieto et al. (2023) shows that training SSMs is brittle in terms of various hyperparameters. Many convolutional models have been proposed for sequence modelling, see e.g. Fu et al. (2023); Li et al. (2022); Shi et al. (2023a). These works parameterize the convolution kernels with specific structures. The Hyena architecture was proposed in Poli et al. (2023) and distilling it into an SSM was studied in Massaroli et al. (2024). Other proposed convo-

lutional models include the LongConv (Fu et al., 2023) and SGConv (Li et al., 2022) architectures, as well as multi-resolution convolutional models (Shi et al., 2023b).

**Spectral filtering.** A promising technique for learning in linear dynamical systems with long memory is called spectral filtering put forth in Hazan et al. (2017). This work studies online prediction of the sequence of observations $y_t$, and the goal is to predict as well as the best symmetric LDS using past inputs and observations. Directly learning the dynamics is a non-convex optimization problem, and spectral filtering is developed as an improper learning technique with an efficient, polynomial-time algorithm and near-optimal regret guarantees. Different from regression-based methods that aim to identify the system dynamics, spectral filtering's guarantee does not depend on the stability of the underlying system, and is the first method to obtain condition number-free regret guarantees for the MIMO setting. Extension to asymmetric dynamical systems was further studied in Hazan et al. (2018). Spectral filtering is particularly relevant to this study since it is a convolutional model with fixed filters. Thus, our results can be immediately applied to this technique and imply provable regret bounds with guaranteed running time, improving upon the state of the art.

**Online learning and regret minimization in sequence prediction.** The methodology of online convex optimization, see e.g. Hazan et al. (2016), applies to sequences prediction naturally. In this setting, a learner iteratively predicts, and suffers a loss according to an adversarially chosen loss function. Since nature is assumed to be adversarial, statistical guarantees are not applicable, and performance is measured in terms of regret, or the difference between the total loss and that of the best algorithm in hindsight from a class of predictors. This is a particulary useful setting for sequential prediction since it requires no assumptions on the true sequence and leads to robust methods. Sequential prediction methods that apply to dynamical systems are more complex as they incorporate the notion of a state. Recently the theory of online convex optimization has been applied to learning in dynamical systems, and the spectral filtering methodology was developed in this context. See Hazan & Singh (2022) for an introduction to this area.

In independent work Oncescu et al. (2024) presents a very similar algorithm for inference with convolutional models, with a total runtime of $O(L \log^2(L))$ (same as our Continuous-FutureFill result) via the method of relaxed polynomial interpolation. Our algorithm builds on the simple and intuitive idea of FutureFill, allowing us to create a spectrum of trade-offs between compute and memory. An intermediate point on this spectrum is the Epoched-FutureFill algorithm, which has a streamlined implementation, low memory usage, and potentially stronger performance in practice.

### D.2 MORE DETAILS ON CONVOLUTIONAL SEQUENCE PREDICTION MODELS

**State Space Models** State space models such as those considered in Gu et al. (2021a) have shown considerable success and adoption for long range sequence modelling. They can be defined via the following dynamics equation of a Linear Dynamical System (LDS)

$$x_t = Ax_{t-1} + Bu_t, y_t = Cx_t + Du_t \tag{2}$$

where $u, y$ are the input and output sequences and $A, B, C, D$ are the learned parameters. Various works deal with specifications of this model including initialization (Gu et al., 2020), diagonal versions (Gupta et al., 2022), gating (Mehta et al., 2023) and other effective simplifications (Smith et al., 2023). All these models can be captured by convolutional models since the output sequence $y$ in equation 2 can be written as

$$y = \phi * u + Du,$$

where the filter $\phi$ satisfies $\phi_i = CA^{i-1}B$. Thus a convolutional sequence model with learnable filters $\phi$ generalizes these SSMs. However, SSMs are more efficient for generation as they can generate a token in constant time.

**LongConv/SGConv.** The LongConv (Fu et al., 2023) and SGConv (Li et al., 2022) architectures exploit the above connection and propose direct regularizations of the convolution kernel to bias them towards representing a state space model.

**Spectral Transform Units.** The STU architecture was proposed in Agarwal et al. (2023) based on the spectral filtering technique for linear dynamical systems (Hazan et al., 2017; 2018). These are

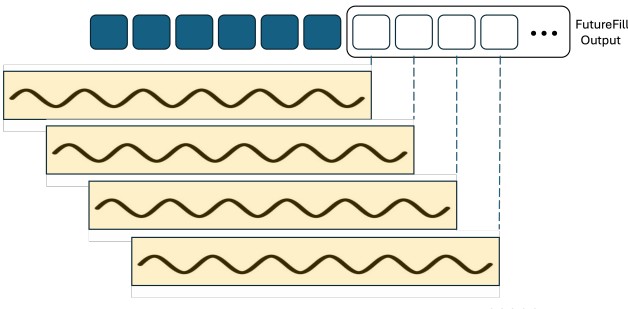

Figure 5: FutureFill between an input sequence and a convolutional filter.

convolutional sequence models based on carefully constructed filters that are **not data-dependent**. Let $\phi_1, ..., \phi_k$ be the first $k$ eigenvectors of the Hankel matrix $H_L$ given by

$$H_L = \int_0^1 \mu_\alpha \mu_\alpha^\top d\alpha \ \in \mathbb{R}^{L \times L}, \quad \mu_\alpha = (\alpha - 1)[1, \alpha, \alpha^2, .., \alpha^{L-1}].$$

The STU predicts according to the following rule [5] $\hat{y}_t = \sum_{i=1}^k M_i \langle \phi_i, u_{t:t-L} \rangle$, where $M_{1:k}$ are learned projection matrices. Note that the inner products $\langle \phi_i, u_{t:t-L} \rangle$ are the outputs of $\phi_i * u$. The STU architecture is particularly appealing for learning LDS with long memory, as demonstrated by its dimension-free sublinear regret guarantees for this setting (Agarwal et al., 2023).

### D.3 ALGORITHM SCHEMATICS

We provide illustrations of the FutureFill operation in Figure 5. We further provide schematics describing our Algorithms 1 and 2 in Figures 6 and 7

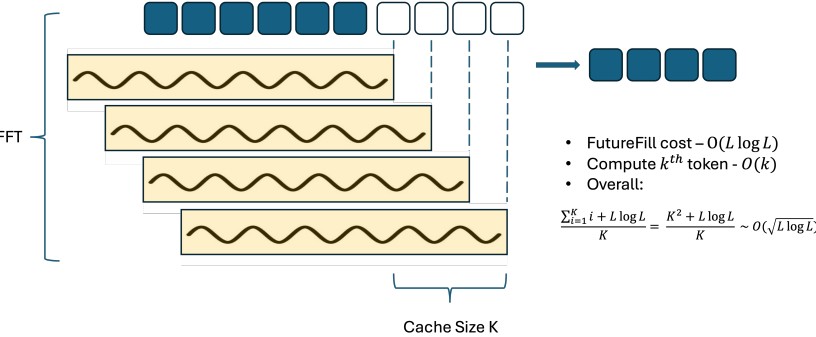

Figure 6: Illustration for Algorithm 1

### D.4 ALGORITHM FOR FAST AUTO-REGRESSIVE SEQUENCE GENERATION FROM A PROMPT

## E FAST ONLINE CONVOLUTIONAL PREDICTION

In this section, we give a more detailed treatment on how FutureFill improves online convolutional prediction in the context of regret minimization. When predicting a sequence in an auto-regressive fashion, an online learner iteratively sees an input $u_t$ and has to predict output $\hat{y}_t$, after which the true output $y_t$ is revealed. The goal is to minimize error according to a given Lipschitz loss function

---

[5]more precisely, there are additional linear and constant terms depending on the exact filters used, such as $\hat{y}_t = \hat{y}_{t-2} + \sum_{i=1}^3 M_i^u u_{t+1-i} + \sum_{i=1}^k M_i \langle \phi_i, u_{t:t-L} \rangle$, see Agarwal et al. (2023) for more details.

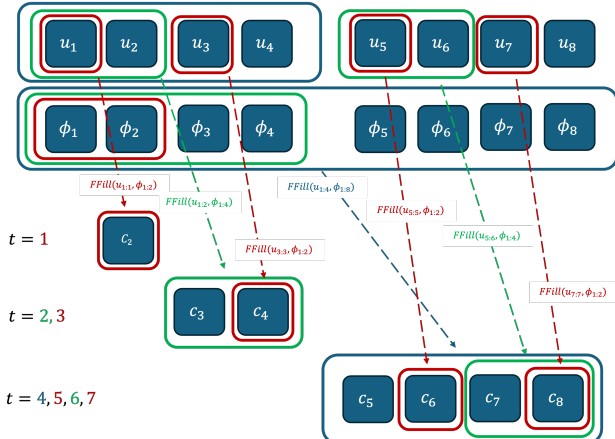

Figure 7: Quasilinear Online Convolution using FutureFill: Figure shows the execution flow for Algorithm 2 for convolving 8-length sequences. Input sequence $u$ streams in an online fashion and filter $\phi$ is fully available to the algorithm. Colors are representative of the size of the FutureFill operations performed and the time $t$ (also color-coded) highlights when the FutureFill operations were performed.

---

**Algorithm 3** Fast auto-regressive sequence generation from a prompt using FutureFill

1: **Input:** Generation length $K > 0, L > 0$, prompt $p_{1:L}$, convolutional filter $\phi \in \mathbb{R}^{L+K}$.
2: Set up a FutureFill cache $C \in \mathbb{R}^K$ as $C \leftarrow \text{FutureFill}(p, \phi)$.
3: Set up the online convolution algorithm with filter $\phi$ and sequence length $K$, i.e. $\mathcal{A} \leftarrow \text{ContinuousFutureFill}(\phi)$.
4: Running candidate token $y \leftarrow 0$.
5: **for** $t = 1, ..., K$ **do**
6:     Output $\hat{y}_t = C_t + y$.
7:     Generate next token candidate $y \leftarrow \mathcal{A}(\hat{y}_t)$.
8: **end for**

---

$\ell_t(y_t, \hat{y}_t)$. In online learning it is uncommon to assume that the true output sequence was generated by the same family of models as those learned by the learner. As a result the metric of performance is usually taken to be regret. Given a class of possible predictors, the goal is to minimize regret with respect to these predictors. For example, a linear predictor predicts according to the rule

$$\pi_{M_{1:k}, N_{1:l}}(u_{1:t}, y_{1:t-1}) = \sum_{i=1}^{k} M_i u_{t-i} + \sum_{j=1}^{l} N_j y_{t-j}.$$

The performance of a prediction algorithm $\mathcal{A}$ is measured by regret, or difference in total loss compared to a class of predictors $\prod$, such as that of linear predictors, e.g.

$$\text{Regret}_T(\mathcal{A}) = \sum_{t=1}^{T} \ell_t(y_t, \hat{y}_t^{\mathcal{A}}) - \min_{\pi \in \prod} \sum_{t=1}^{T} \ell_t(y_t, \hat{y}_t^{\pi}).$$

This formulation is valid for online sequence prediction of any signal. We are particularly interested in signals that are generated by dynamical systems. A partially observed time-invariant linear dynamical system is given by the dynamics equations

$$x_{t+1} = Ax_t + Bu_t + w_t, \quad y_t = Cx_t + Du_t + \zeta_t,$$

where $x_t$ is the (hidden) state, $u_t$ is the input or control to the system, and $y_t$ is the observation. The terms $w_t, \zeta_t$ are noise terms, and the matrices $A, B, C, D$ are called the system matrices. A linear dynamical predictor with parameters $A, B, C, D$ predicts according to

$$\pi_{ABCD}(u_{1:t}, y_{1:t-1}) = \sum_{i=1}^{t-1} CA^{i-1} Bu_{t-i} + Du_t.$$

The best such predictor for a given sequence is also called the optimal open loop predictor, and it is accurate if the signal is generated by an LDS without noise.

When modeling long-range dependencies, the class of marginally stable linear dynamical systems is of particular interest. Marginally stable systems are systems whose dynamics matrix $A$ has eigenvalues of magnitude up to 1, and thus observations $y_t$ can depend on inputs that are arbitrarily far in the past. The long-range dependencies also make learning these systems challenging, and most techniques based on system identification do not have guarantees in this setting. The spectral filtering algorithm (Hazan et al., 2017) is a convex relaxation of the problem of learning marginally stable LDS online, and was the first algorithm to achieve sublinear, hidden dimension-free regret for learning systems with symmetric dynamics matrices. Spectral filtering uses convolutions to compute the prediction at each time step, and we demonstrate below how FutureFill can naturally be applied to accelerate this algorithm.

### E.1 CASE STUDY: FAST ONLINE SPECTRAL FILTERING

We illustrate in more detail how the method works for the spectral filtering algorithm from Hazan et al. (2017), improving the total running time from $O(L^2)$ to $O(L \log^2 L)$ while maintaining the same regret bound.

---

**Algorithm 4** Efficient Spectral Filtering via FutureFill

---

1: **Input:** Number of filters $N > 0, L > 0$.
2: Set variables $\{M_1^1 \ldots M_N^1 \in \mathbb{R}^{d_{out} \times d_{in}} \leftarrow 0\}$ and set $\{\phi_1 \ldots \phi_N\}$ as the largest eigenvectors of $H_L$, the Hankel matrix corresponding to length-$L$ sequences.
3: Initialize $N$ OnlineConvolution modules, one for each filter $\{\mathcal{A}_k(\phi_k)\}_{k=1}^N$.
4: **for** $t = 1, 2, ..., L$ **do**
5:     Receive input token $u_t$.
6:     **for** $k = 1, 2, \ldots N$ **do**
7:         $F_k \leftarrow \mathcal{A}_k(\phi_k)(u_t)$.
8:     **end for**
9:     Compute and predict $\hat{y}_t = \sum_{k=1}^N M_k^t F_k$.
10:     Observe $y_t$, suffer loss $\ell_t(M_{1:k}^t) = \|y_t - \hat{y}_t\|^2$, and update $M_{1:k}^{t+1} \leftarrow \nabla \ell_t(M_{1:k}^t)$.
11: **end for**

---

The main claim regarding the performance of Algorithm 4 follows directly from Theorems 2 and 3 and is as follows.

**Corollary 7.** *Algorithm 4 with sequence length $L$ guarantees the same regret bound as spectral filtering (Hazan et al., 2017) with context length $L$. Furthermore its computational complexity based on the online convolution module used are as follows:*

- *If using EpochedFutureFill(Algorithm 1): Runtime - $O(L^{3/2}\sqrt{\log L})$, Memory - $O(\sqrt{L \log L})$.*

- *If using ContinuousFutureFill(Algorithm 2): Runtime - $O(L \log^2 L)$, Memory - $O(L)$.*

## F DEFERRED PROOFS

*Proof of Proposition 1.* Note that by definition, $[a * b]_s = \sum_{i=1}^s a_i b_{s+1-i}$. We now consider the two cases: for $s \leq t_1$, we have that

$$[a_{1:t_1} * b_{1:t_1}]_s = \sum_{i=1}^s a_i b_{s+1-i} = [a * b]_s.$$

For the case when $t \geq s > t_1$, we have that

$$[a_{t_1+1:t} * b_{1:t-t_1}]_{s-t_1} = \sum_{i=1}^{s-t_1} a_{t_1+i} b_{s-t_1+1-i} = \sum_{i=t_1+1}^s a_i b_{s+1-i},$$

where the last equality follows by redefining $i = t_1 + i$. Further we have that

$$[\text{FutureFill}(a_{1:t_1}, b)]_{s-t_1} = \sum_{i=1}^{t-s+t_1} a_{t_1-i+1} \cdot b_{s-t_1+i} = \sum_{i=1}^{t_1} a_{t_1-i+1} \cdot b_{s-t_1+i} = \sum_{i=1}^{t_1} a_i \cdot b_{s+1-i},$$

where the second last equality follows by noting that $a_j$ is assumed to be 0 for all $j \leq 0$ and the last equality follows by redefining $i = t_1 - i + 1$. Overall putting the two together we get that

$$[a_{t_1+1:t} * b_{1:t-t_1}]_{s-t_1} + [\text{FutureFill}(a_{1:t_1}, b)]_{s-t_1} = \sum_{i=1}^{t_1} a_i \cdot b_{s+1-i} + \sum_{i=1}^{t_1} a_i \cdot b_{s+1-i} = \sum_{i=1}^{s} a_i \cdot b_{s+1-i} = [a*b]_s.$$

This finishes the proof. $\qquad\square$

### F.1 Proofs for Algorithm 1

*Proof of correctness for Algorithm 1.* Consider any time $t$ and the output $\hat{y}_t$. Let $t' \leq t$ be the last time when Line 6 was executed, i.e. FutureFill was computed. By definition $t' = t - \tau$. Note the following computations.

$$\hat{y}_t = \sum_{j=1}^{\tau} u_{t+1-j} \cdot \phi_j + C_\tau = \sum_{j=1}^{\tau} u_{t+1-j} \cdot \phi_j + [\text{FutureFill}(u_{1:t'}, \phi_{1:t'+K})]_\tau$$

$$= \sum_{j=1}^{\tau} u_{t+1-j} \cdot \phi_j + \sum_{j=1}^{t'+K-\tau} u_{t'-j+1} \cdot \phi_{\tau+j}$$

$$= \sum_{j=1}^{\tau} u_{t+1-j} \cdot \phi_j + \sum_{j=1}^{t'} u_{t'-j+1} \cdot \phi_{\tau+j}$$

$$= \sum_{j=1}^{\tau} u_{t+1-j} \cdot \phi_j + \sum_{j=1}^{t-\tau} u_{t-\tau-j+1} \cdot \phi_{\tau+j}$$

$$= \sum_{j=1}^{\tau} u_{t+1-j} \cdot \phi_j + \sum_{j=\tau+1}^{t} u_{t-j+1} \cdot \phi_j = [u * \phi]_t$$

$\qquad\square$

### F.2 Proofs for Algorithm 2

*Proof of Theorem 3.* As can be seen from the algorithm for every generated token the most expensive operation is the FutureFill computed in Line 6 so we bound the total runtime of that operation. Note that at any time $t$, the cost of FutureFill operation is $O((1 \vee k(t)) \cdot 2^{k(t)})$, where $a \vee b$ denotes the max of $a$ and $b$. Summing this over every time step $t$ we get,

$$\sum_{t=1}^{L} (1 \vee k(t)) 2^{k(t)} = \sum_{k=0}^{\lfloor \log L \rfloor} |\{t : k(t) = k\}| (1 \vee k) 2^k$$

$$\leq L + \sum_{k=1}^{\lfloor \log L \rfloor} 2^{\lfloor \log L \rfloor - k + 1} \cdot k 2^k \leq 3L \sum_{k=1}^{\lfloor \log L \rfloor} k \leq 3L \log^2 L.$$

Thus the total runtime of the algorithm is bounded by $O(L \log^2 L)$. $\qquad\square$

*Proof of correctness for Algorithm 2.* We will focus on showing that $C_t = \sum_{i=2}^{t} u_{t+1-i} \phi_i$. Since the output is $C_t + u_t \cdot \phi_1$, this will suffice for the proof. For brevity of the proof and without loss

of generality we will assume $L$ is a power of 2. For cleaner presentation for the $s^{th}$ coordinate of vector $v$ we will use the notation $v_s$ and $v[s]$ interchanegably in this section.

We first introduce some definitions for convenience in this section. Given an index $i \leq L$ we define its decomposition $\{i_1, i_2 \ldots i_m\}$ as the unique sequence of numbers $\leq \log L$ such that following holds

$$i_1 > i_2 > i_3 \ldots \text{ and } i = \sum_j 2^{i_j}.$$

These indices correspond to the ones in a $\log L$-bit representation of $i$. Note that $k(i)$ as defined in the algorithm is equal to $i_m$. Further we define the cumulants of $i$ as the following sequence of numbers $\{i'_1, i'_2 \ldots\}$ satisfying

$$i'_\tau = \sum_{j=1}^{\tau} 2^{i_j}.$$

Thus we have that $i'_1 < i'_2 < \ldots i'_m = i$. We now prove the following lemma which specifies when the FutureFill cache gets updated in an execution of the algorithm.

**Lemma 8.** *Given an index $i \leq L$, consider its decomposition $\{i_1, i_2 \ldots i_m\}$ and cumulants $\{i'_1, i'_2 \ldots i'_m\}$ as defined above. It holds that the value of $C_{i+1}$ is updated (as in Line 8 in the algorithm) only when $t$ is one of $\{i'_1, i'_2 \ldots i'_m\}$.*

A direct consequence of the above lemma is that given any index $i$ we have that the value of $C_{i+1}$ is not updated after time step $i$. Further using the decomposition $\{i_1, i_2 \ldots i_m\}$ and cumulants $\{i'_1, i'_2 \ldots i'_m\}$ of $i$ and the update equations for $C$ (Line 8), we have that final value of $C_{i+1}$ is given by the following,

$$C_{i+1} = \sum_{j=1}^{m} \text{FutureFill}(u[i'_j - 2^{i_j} + 1 : i'_j], \phi[1 : 2^{i_j+1}])[i + 1 - i'_j]$$

$$= \sum_{j=1}^{m} \sum_{k=1}^{2^{i_j}} u[i'_j - k + 1] \cdot \phi[i + 1 - i'_j + k]$$

$$= \sum_{j=1}^{m} \sum_{r=i'_j - 2^{i_j}+1}^{i'_j} u[r] \cdot \phi[i + 1 - r + 1]$$

$$= \sum_{r=1}^{i} u[r] \cdot \phi[i + 1 - r + 1]$$

Thus the output of the algorithm for any $i$, satisfies

$$\hat{y}_{i+1} = C_{i+1} + u_{i+1} \cdot \phi_1 = \sum_{r=1}^{i} u[r] \cdot \phi[i+1-r+1] + u_{i+1} \cdot \phi_1 = \sum_{r=1}^{i+1} u[r] \cdot \phi[i+1-r+1] = [u * \phi]_{i+1}.$$

This proves the requisite. We finally provide a proof of Lemma 8 to finish the proof.

*Proof of Lemma 8.* By the definition of the algorithm, to be able to update $C_{i+1}$ at some time $t < i + 1$ it must be the case that
$$i + 1 \in [t + 1, t + 2^{k(t)}].$$

Consider some $t$ and its decomposition $\{t_1, t_2 \ldots t_n\}$ and cumulants $\{t'_1, t'_2 \ldots t'_n\}$. By the definition of the update in Line 8, we have that at time $t$ we only update indices $i + 1$ for which $i$ has the sequence $\{t'_1, t'_2 \ldots t'_{n-1}\}$ in its decomposition as a prefix. It can then be seen that for a given number $i$, the only such numbers are its cumulants, i.e. $\{i'_1 \ldots i'_m\}$ which finishes the proof. □

□

## G   HYENA ABLATIONS

In this section, we describe evaluations of FutureFill on the Hyena Operator Poli et al. (2023). We consider a Hyena Operator with model dimension 768 and positional encoding MLP dim 33, which are the default parameters in the official Hyena runtime benchmarks, and we evaluate with batch size 1. We maintain all other default parameters. After precomputing the max-length Hyena filters, we test the Hyena Operator on autoregressive generation of outputs of length $2^{14}, 2^{15}, \ldots, 2^{20}$. For FutureFill, we choose $K = \sqrt{2L \log_2 L}$.

| Seq Length | Standardconv (s) | FutureFill (s) | Speedup | Improvement |
|---|---|---|---|---|
| 16,384 | 1.368 | 1.492 | 0.917x | -9.1% |
| 32,768 | 4.483 | 3.860 | 1.161x | 13.9% |
| 65,536 | 16.295 | 12.463 | 1.307x | 23.5% |
| 131,072 | 62.436 | 45.902 | 1.360x | 26.5% |
| 262,144 | 247.990 | 179.254 | 1.383x | 27.7% |
| 524,288 | 997.250 | 720.268 | 1.385x | 27.8% |
| 1,048,576 | 4027.779 | 2928.229 | 1.375x | 27.3% |

Table 19: Performance comparison for autoregressive generation for a Hyena Operator with standard convolutions and Hyena Operator with FutureFill. Generation time is measured in seconds.

