# OpenReview forum: "FutureFill: Fast Generation from Convolutional Sequence Models"
_ICLR.cc/2026/Conference — ICLR 2026 Poster_

### Official Review · Reviewer_y912 · 2025-10-23

**Soundness:** 3
**Presentation:** 3
**Contribution:** 3
**Rating:** 6
**Confidence:** 2

**Summary:**

This paper continues the line of work aiming to speed up inference with respect to sequence length and to avoid the quadratic complexity of standard Transformers. The authors propose a method for fast generation using convolutional sequence models. They focus on exact auto-regressive generation from convolutional models, reducing both generation time and cache size, achieving O(N log N) complexity instead of O(L²).
Empirical results at small scale show that the proposed algorithms achieve sub-quadratic scaling compared to naive convolution implementations, and they report up to 1.7× speedup over the baseline.
Overall, this is a solid  paper that provides a practical speedup method, but the experimental validation feels limited.

**Strengths:**

The work is technically sound and contributes to the ongoing effort of making sequence models more efficient at inference time.

**Weaknesses:**

The experiments are limited to relatively small language models (below 1B parameters), which makes it hard to assess the impact at realistic scales. In addition, only inference speed results are presented — there is no evaluation of model quality (e.g., perplexity or downstream performance), which is important to verify that the gains do not come at the cost of degraded output quality.

**Questions:**

-

---

> ### Author Response · Authors · 2025-11-22
>
> Dear Reviewer,
>
> We are grateful for the recognition of the soundness and relevance of our contribution. We address the concerns below:
>
> **[Scalability to multibillion parameter models]** Our current resource constraints do not allow us to run experiments on models with 1B or more parameters. However, larger versions of convolutional models, such as FlashSTU and Hyena, often increase parameters through further layers or greater residual dimension, which corresponds to more convolutions of similar length and greater batch size. Our experiments in Appendix B extensively ablate the impact of layers and residual dimension, and as our contribution is with a convolution algorithm, the scaling is well understood and approximately linear in both parameters.
>
> **[No Evaluation on Model Quality]** Thank you for the question. We show theoretically that FutureFill and the baseline naive convolution are mathematically equivalent for decoding. FutureFill is an exact decoding algorithm and does not fundamentally alter model quality. We verify this experimentally in Appendix B.6, where we report results of language modelling benchmarks using the 330M parameter pretrained hybrid-STU model, using bfloat16 weights (and upcasting the FFT to float16 or float32). Our experiments show that FutureFill does not degrade accuracy. In our implementation, we find only exceptionally minor differences between the naive convolution and FutureFill, entirely due to numerical precision.
>
> Thank you for the questions, and we agree that a crucial benefit of FutureFill is that it comes at no cost to model accuracy or output quality. In our revision, we will place added emphasis on the stability of FutureFill and its impact on model quality. We would be more than happy to elaborate on any additional concerns you have with the paper, and we appreciate your time and effort.
>
> Thank you,
>
> Authors

---

> > ### Comment · Reviewer_y912 · 2025-11-24
> > **acknowledge reviewers' answer**
> >
> > I acknowledge reviewers' response and thanks them for their answers.
> > Thanks for pointing to Appendix B.6 about related to model quality.
> > All in all this confirms my good/positive opinion about the paper.

---

### Official Review · Reviewer_d6Cp · 2025-10-30

**Soundness:** 4
**Presentation:** 4
**Contribution:** 3
**Rating:** 8
**Confidence:** 3

**Summary:**

This paper introduces "FutureFill," a novel and efficient method for auto-regressive generation from sequence models that use convolutional operators. The primary contribution is an algorithmic technique that reduces the computational complexity of generating L tokens from scratch from a quadratic $O(L^2)$, which is typical for naive online convolution, to a quasilinear. The paper presents two concrete algorithms based on this idea:

1. Continuous-FutureFill: An algorithm that achieves the $O(L log^2 L)$ runtime with $O(L)$ memory.
2. Epoched-FutureFill: A more practical variant that offers a trade-off between runtime and memory.

Furthermore, the paper shows that when generating K tokens from a prompt of length L, FutureFill significantly reduces the required cache size from $O(L+K)$ to $O(K)$, a crucial improvement for long-context applications. The authors validate their theoretical claims with experiments on both synthetic data and large-scale (up to 826M parameters) convolutional language models, demonstrating empirical speedups of up to 2x over baseline methods on modern hardware.

**Strengths:**

1: The paper tackles a critical and well-known bottleneck in sequence modeling, i.e. the slow quadratic-time generation process for models based on convolutions. Making this efficient, especially in long-sequnece scenerios is a major practical contribution.

2. The paper is clearly written and the theoretical claims are well-supported by theorems and complexity. The experiments not only demonstrate asymptotic behavior but also wall-clock time improvements.

3. The idea of futurefill is intuitive and based on observations and intuition regarding the properties of convolution and FFT.

**Weaknesses:**

1. The paper points out that there has been independent and concurrent work that achieves the same runtime complexity, which slightly tempers the novelty.

2. The paper seems to focus only on FlashSTU-T model and would be nice to show how this is generalizable to other convolutional models (e.g. Hyena).

3. While the paper does demonstrate real speed improvement of 2x, it was not as dramatic as the difference from $O(L^2)$ to $O(L)$. More detailed analysis here on why the speedup was not significant would be interesting and informative.

**Questions:**

1. Compared to other works that achieve the same complexity, could the authors elaborate more on the qualitative or practical differences between FutureFill and other concurrent works?

2. How would the algorithm perform if the generation length is not known in advance, i.e. the epoch length
 was set to the theoretical optimum but what should peopole use in real-world scenerios?

---

> ### Author Response · Authors · 2025-11-22
>
> Dear Reviewer,
>
> Thank you for your time and for your feedback.  We address each concerns below:
>
> **[Comparison with Oncescu et al.]** Compared with Oncescu et al., our work has three primary differences:
> - Our approach is flexible and offers a spectrum of tradeoffs between runtime and memory. We give two points on this spectrum – EpochedFF and ContinuousFF. Oncescu et al. proposes an algorithm similar to the ContinuousFF variant.
> - Compared to ContinuousFF, EpochedFF requires less memory ($\tilde{O}(\sqrt{L})$ vs. $O(L)$)  and has a slower runtime ($\tilde{O}(L^{3/2})$ vs. $\tilde{O}(L)$). This variant is useful when the user is memory-constrained, it is also much simpler to implement and makes less calls to the FFT, the fixed costs of which might outweigh the benefits of faster runtime.
> - We derive FutureFill from the perspective of online learning, while Oncescu et al. frames the problem as relaxed polynomial interpolation.
>
> Within the paper we emphasize different attributes, with FutureFill offering extensive ablations across model depth, residual dimension, and prefill length with the FlashSTU model, whereas Oncescu predominantly focuses on the Hyena model.
>
> **[Hyena Experiments]** To showcase the improvement of FutureFill on other convolutional models, we extend our experiments to the popular Hyena model. We benchmarked FutureFill compared to the naive online convolution baseline on a Hyena model with model_dim=768, a single operator, and batch_size = 1, using the default parameters in the Hyena runtime tests[1]. Compared to the baseline, FutureFill provides a ~1.4x speedup on the longest sequences (131,072 to 1,048,576), and we have included the table of results below. For the shorter sequence lengths, the GPU is underutilized and convolution is not compute bound, limiting the performance improvement. Full details are available in Appendix G.
>
> | Sequence Length | 16,384 | 32,768 | 65,536 | 131,072 | 262,144 | 524,288 | 1,048,576 |
> | --- | --- | --- | --- | --- | --- | --- | --- |
> | Standardconv (s) | `1.368` | `4.483` | `16.295` | `62.436` | `247.990` | `997.250` | `4027.779` |
> | FutureFill (s) | `1.492` | `3.860` | `12.463` | `45.902` | `179.254` | `720.268` | `2928.229` |
> | Speedup  | `x0.917` | `x1.161` | `x1.307` | `x1.360` | `x1.383` | `x1.385` | `x1.375` |
>
> [1] Poli & al, Hyena Hierarchy: Towards Larger Convolutional Language Models
>
> **[Real Speed Improvement]** For the isolated online convolutional experiments (Figure 1), we notice close to the predicted scaling trends, although achieving an approximately ~2x improvement requires 1e6 generation length since the convolution is a well-optimized operation on any accelerated hardware, and often overhead costs of memory transfers and kernel launching can dominate runtime especially at lower sequence lengths. With FlashSTU-T, we similarly have that other non-arithmetic costs can be dominant until sufficient sequence lengths, and due to compute limits, we were unable to test exceptionally long sequence-lengths.
>
> **[Practical Setup]** In most practical setups, the max prompt length and the max generations are specified. In these cases we can use the max lengths to set the epoch length. If they are not given, we can use the doubling trick: we can first collect historical data on the typical generation length T, and proceed as if we will generate T tokens. If we need to generate more tokens, we increase the planned generation length to 2T, and change the epoch length accordingly. We can double the planned generation length every time we reach the end of the current generation length. This trick ensures that the total runtime is within factor 2 of the optimal runtime for EpochedFF.
>
> We appreciate the detailed questions, and we hope these responses clarify the paper!

---

> > ### Comment · Reviewer_d6Cp · 2025-11-26
> >
> > Thank you for the detailed reply. I will keep my score.

---

### Official Review · Reviewer_9QkU · 2025-10-31

**Soundness:** 4
**Presentation:** 4
**Contribution:** 4
**Rating:** 6
**Confidence:** 2

**Summary:**

The paper positions itself with respect to recent work that uses convolutional operators as a way of mitigating complexity issues in autoregressive attention-based models.  Specifically, they propose a method FutureFill that reduces complexity in text generation below quadratic in context length; a core part of the idea is that there is a memory trade-off that permits the reduction in inference time complexity, which also allows a spectrum of algorithm variants in terms of that trade-off.  The paper contains some theoretical results on this complexity, and also some experimental results looking empirically at inference times as a function of context length, as well as checking the performance on several downstream tasks.

**Strengths:**

* Overall, I think this is quite a strong paper.  The use of convolutional approaches as in e.g. Hyena is an important direction to address the quadratic complexity issue, and this paper’s contribution looks like an important step in that, and could well be adopted quite widely as a source of performance improvements.

**Weaknesses:**

These are mostly relatively minor.

* The Abstract is fairly short and bare-bones; it’s not really until reading the paper that the actual importance of the work comes through.

* It’s fairly reasonable given space constraints to save most of the literature review of Sec 1.1 for the appendix.  However, something that I thought was missing in both Sec 1.1 and the appendix’s extended version was the discussion of differences wrt Oncescu et al. (2024).  This is presented as an independent work that achieves the same complexity result as the present paper, but there’s no argument made as to why FutureFill then is necessary.  Is there something more advantageous about the memory trade-off inherent in FutureFill?  Is there some limitation to Oncescu?  It’s not made clear why a new method with this complexity is a useful thing to have, and this is quite crucial for understanding the importance of the present paper's contribution.

* Fig 5 in App D.3 is presented a depiction of the FutureFill operation between an input sequence and a convolutional filter; it’s where the reader is supposed to get a concrete idea of how the method works, to supplement the mathematical definitions.  However, the diagram is very schematic and abstract, and not actually much of a help.  Some fleshing out of the diagram and explanation in D.3 would be helpful.

* For the proof of Propn 1, it would be helpful to link explicitly to App F.

**Questions:**

Please see above.

---

> ### Author Response · Authors · 2025-11-22
>
> Dear Reviewer,
>
> Thank you for your time and effort reviewing the paper, and we appreciate the positive feedback. We will address the weaknesses below.
>
> **[Comparison with Oncescu et al.]** Compared with Oncescu et al., our work has three primary differences:
> - Our approach is flexible and offers a spectrum of tradeoffs between runtime and memory. We give two points on this spectrum – EpochedFF and ContinuousFF. Oncescu et al. proposes an algorithm similar to the ContinuousFF variant.
> - Compared to ContinuousFF, EpochedFF requires less memory ($\tilde{O}(\sqrt{L})$ vs. $O(L)$)  and has a slower runtime ($\tilde{O}(L^{3/2}$) vs. $\tilde{O}(L)$). This variant is useful when the user is memory-constrained, and it is also much simpler to implement.
> - We derive FutureFill from the perspective of online learning, while Oncescu et al. frames the problem as relaxed polynomial interpolation.
>
> Within the paper we emphasize different attributes, with FutureFill offering extensive ablations across model depth, residual dimension, and prefill length with the FlashSTU model, whereas Oncescu predominantly focuses on the Hyena model.
>
> **[Abstract]** We provide an improved abstract that better contextualizes the benefits of FutureFill:
>
>
> “In response to the quadratic generation time of Transformed-based models, recent research in sequence-to-sequence models has focused on alternative architectures based on the convolution primitive. We address the challenge of efficient auto-regressive generation by introducing FutureFill—a general-purpose, generation method for sequence-to-sequence prediction algorithms based on convolutional operators. FutureFill reduces autoregressive generation time from quadratic to quasilinear in the context length. Moreover, when generating from a prompt, it maintains a cache whose size grows only with the number of tokens to be generated—often much smaller than the caches required by standard convolutional or attention-based models. We validate our theoretical claims with experiments on synthetic tasks and demonstrate substantial efficiency gains when generating from a deep convolutional sequence prediction model.”
>
> **[Fig 5 in App D.3]** Thank you for the feedback, we are working on an improved schematic that shows the convolution to build the cache (i.e., close to current Fig 5) and how the cache combines with the shorter convolution to produce the output. We agree the current Fig. 5 is confusing, especially given that it is not clear from the figure how the cache and online contribution interact, and since the figure does not explicitly reference the cache.
>
> **[Proof of Propn 1]** Thank you for catching this, we will add the link.

---

### Official Review · Reviewer_SKar · 2025-11-02

**Soundness:** 3
**Presentation:** 3
**Contribution:** 3
**Rating:** 6
**Confidence:** 3

**Summary:**

The paper introduces FutureFill, a fast inference algorithm for convolutional sequence models that reduces generation complexity by precomputing future contributions via FFT-based convolutions. The method maintains exact output equivalence with standard convolution while achieving notable latency improvements on FlashSTU-T models, all without retraining or architectural changes. It also presents two practical variants—Epoched and Continuous FutureFill—that balance memory and speed for different deployment settings.

**Strengths:**

- Addresses a less-studied yet important bottleneck in convolutional language models.
- Strong theoretical foundation with clear runtime and correctness guarantees.
- Training-free and exact—no compromise on model quality.
- Shows consistent practical gains and integrates seamlessly with existing architectures.
- Provides clear implementation details enabling reproducibility.

**Weaknesses:**

- Scalability to multi-billion parameter models not yet validated.
- Baselines limited; comparison with Hyena, RWKV, and S4 models would be useful.
- Reports only latency metrics; including FLOPs or energy-based analysis would make results more hardware-agnostic.
- Hardware dependency unclear—speedups may vary across GPUs/TPUs.
- Memory–latency trade-offs and cache behavior could be analyzed more deeply.

**Questions:**

- How does FutureFill scale with model size and longer sequence lengths (e.g., >100K tokens)?
- Are the latency gains consistent across different hardware backends?
- Could the authors report FLOPs per token to complement latency results?
- How does the method compare with other efficient convolutional architectures in total throughput and memory cost?

---

> ### Author Response · Authors · 2025-11-22
>
> Thank you for your time and for your constructive feedback on the paper. We appreciate your positive assessment of the practical improvements and theoretical contributions of our work. We address each of the weaknesses and questions below:
>
> **[Scalability to multibillion parameter models]** Our current resource constraints do not allow us to run experiments on models with 1B or more parameters. However, larger versions of convolutional models, such as FlashSTU and Hyena, often increase parameters through further layers or greater residual dimension, which corresponds to more convolutions of similar length and greater batch size. Our experiments in Appendix B extensively ablate the impact of layers and residual dimension, and as our contribution is with a convolution algorithm, the scaling is well understood and approximately linear in both parameters.
>
> **[Baselines limited]** To showcase the impact of FutureFill on other convolutional models, we extend our experiments to the popular Hyena model [1]. We benchmarked FutureFill compared to the online convolution baseline on a Hyena model with model_dim=768, a single operator, and batch_size = 1, which are the default parameters in the Hyena runtime tests. Compared to the baseline, FutureFill provides a ~1.4x speedup on the longest sequences (131,072 to 1,048,576), and we have included the table of results below. For the shorter sequence lengths, the GPU is underutilized and convolution is not compute bound, limiting the performance improvement. Full details are available in Appendix G.
>
> | Sequence Length | 16,384 | 32,768 | 65,536 | 131,072 | 262,144 | 524,288 | 1,048,576 |
> | --- | --- | --- | --- | --- | --- | --- | --- |
> | Standardconv (s) | `1.368` | `4.483` | `16.295` | `62.436` | `247.990` | `997.250` | `4027.779` |
> | FutureFill (s) | `1.492` | `3.860` | `12.463` | `45.902` | `179.254` | `720.268` | `2928.229` |
> | Speedup  | `x0.917` | `x1.161` | `x1.307` | `x1.360` | `x1.383` | `x1.385` | `x1.375` |
>
> [1] Poli & al, Hyena Hierarchy: Towards Larger Convolutional Language Models
>
> **[How does FutureFill scale with model size and sequence length]**: For the FlashSTU model, the longest sequence length we could run under our resource constraint was 131k, with the results included in Figure 2(a) and Appendix B. In general, FutureFill’s speedup grows as sequence length increases, across all model sizes. We include extensive ablations across model residual dimension, layer count, and the generation length in Appendix B, showing that the FutureFill benefits extend to all tested configurations. For even longer sequence lengths, we test a basic one-layer convolution with the standard convolution, Epoched FutureFill, and Continuous FutureFill with generation length up to 1e6 in Figure 1, noting that the advantage of FutureFill increases with sequence length.
>
> **[Latency gains across hardware]** To better evaluate the contribution of FutureFill across different hardware, we evaluate the impact of FutureFill on the 670.75M parameters STU-only model on an A100 PCIe (vs H100 SXM in the paper). In the interest of time, we only do 1 warmup run per generation length and then measure the timing on the next run.
> #1 Streaming generation (no prefill)
> This setting is the same as the one displayed in Figure 2.a of the main paper (and associated detailed results in Appendix B.2 - Table 3.)
> | Generation length | 4096 | 8192 | 16384 | 32768 | 65536 | 126976 |
> | --- | --- | --- | --- | --- | --- | --- |
> | FutureFill (in s) | `39.307863` | `82.27319` | `169.5318` | `326.1294` | `734.84865` | `1787.76749` |
> | Baseline (in s) | `32.518088` | `64.5499801` | `137.881302` | `328.02081990` | `954.777773` | `2953.273093` |
> | Speedup | x`0.83` | x`0.78` | x`0.81` | x`1.01` | x`1.30` | x`1.65` |
>
> At the largest generation length Lgen = 126,976, we observe a 1.65 speedup for STU-only, similar to the 1.7 reported in the main paper.
> We note that FutureFill does large FFTs over long sequences. Those benefit a lot from high bandwidth and good cache behavior
> #2 32,768 tokens prompt and generation
> This setting is the same as the one displayed in the results in Table 2 of the main paper (and associated detailed results in Appendix B.3 - Table 6.)
> | Generation length | 4096 | 8192 | 16384 |
> | --- | --- | --- | --- |
> | FutureFill (in s) | `42.2079` | `85.9862` | `170.1594` |
> | Baseline (in s) | `60.91617` | `126.5468` | `273.2210` |
> | Speedup | x`1.44` | x`1.47` | x`1.61` |

---

> ### Author Response · Authors · 2025-11-22
>
> We apologize for the double comment, the experiment table expends many characters in the response.
>
> **[FLOPs per Token]** In addition to latency, we report FLOPs per generated token for our 670.75M parameters STU-only model, with and without FutureFill.
>
> **#1 Streaming generation (no prefill)**
> This setting is associated with Figure 2.a of the main paper (and associated detailed results in Appendix B.2 - Table 3.)
> | Generation length | 4096 | 8192 | 16384 | 32768 | 65536 | 126976 |
> | --- | --- | --- | --- | --- | --- | --- |
> | FutureFill - GFLOPs/ token | `1.3738` | `1.3917` | `1.4406` | `1.5178` | `1.6661` | `1.9928` |
> | Baseline - GFLOPs/ token |  `1.3914` | `1.4417` | `1.5424` | `1.7437` | `2.1464` | `2.9013` |
> | Ratio (Baseline / FutureFill) | `x1.01` | `x1.04` | `x1.07` | `x1.15` | `x1.29` | `x1.46` |
>
> For streaming generation without a prompt, FutureFill reduces FLOPs per token from 2.90 to 1.99 GFLOPs at 128k generated tokens (about 1.5x fewer FLOPs/token).
>
> **#2 32,768 tokens prompt and generation**
> This setting is associated with the results in Table 2 of the main paper (and associated detailed results in Appendix B.3 - Table 6.)
> | Generation length | 4096 | 8192 | 16384 |
> | --- | --- | --- | --- |
> | FutureFill - GFLOPs/ token| `1.3783` | `1.4173` | `1.4666` |
> | Baseline - GFLOPs/ token| `2.1962` | `2.2468` | `2.3476` |
> | Ratio | `x1.59` | `x1.59` | `x1.60` |
> For streaming generation with a prompt of 32,768 tokens, FutureFill reduces FLOPs/token from 2.35 to 1.47 GFLOPs/token for 16k generated tokens (about 1.6x fewer FLOPs/token)
>
> **[Comparison to other efficient convolutional architectures]** Thank you for this question. We want to clarify that we do not propose a new convolution architecture. Instead, we propose an exact decoding algorithm to generate tokens from convolutional architectures. The closest method to our work is Oncescu et al. (Flash Inference) – an independent work. Compared to Oncescu et al., our work has three differences:
> - Our approach is flexible and offers a spectrum of tradeoffs between runtime and memory. We give two points on this spectrum – EpochedFF and ContinuousFF. Oncescu et al. proposes an algorithm similar to the ContinuousFF variant.
> - Compared to ContinuousFF, EpochedFF requires less memory ($\tilde{O}(\sqrt{L})$ vs. $O(L)$)  and has a slower runtime ($\tilde{O}(L^{3/2})$ vs. $\tilde{O}(L)$). This variant is useful when the user is memory-constrained, and it is also much simpler to implement.
> - We derive FutureFill from the perspective of online learning, while Oncescu et al. frames the problem as relaxed polynomial interpolation.
>
> Besides Oncescu et al., Massaroli et al. (Laughing Hyena Distillery) is also motivated by the complexity of generating from convolutional models, and the paper proposes to distill a convolutional model into a state-space model with O(1) inference complexity. Importantly, Massaroli’s approach is approximate, and the approximation gap is not completely understood.
>
> We hope these additional experiments address your concerns, and we would be more than happy to elaborate on any additional concerns you have with the paper.

---

### Meta-Review · Area_Chair_UjJm · 2026-01-06

**Summary:**

The reviewers generally agreed that the paper addresses a critical efficiency bottleneck in non-transformer architectures. The primary concerns revolved around the practical scalability of the method beyond small-scale models (under 1B parameters), the lack of hardware-agnostic metrics, and the distinction from concurrent work. Some reviewers also questioned the generalization of the method to other architectures like Hyena. Through the rebuttal, the authors provided extensive new data and clarifications that largely mitigated these concerns, demonstrating that the speedup is consistent across different architectures and hardware, while maintaining exact mathematical equivalence.

**Reviewer Concerns:**

Most of the concerns are addressed:

- The request for hardware-agnostic analysis was met with new GFLOPs/token reports, showing a 1.5x–1.6x reduction in compute.

- The concern about perplexity/accuracy was resolved by pointing to Appendix.

- Concurrent Work: all questions regarding the con-current work were addressed by highlighting the unique memory-runtime trade-off and its derivation from online learning of the method.

**Reviewer Scores:**

All reviewers hold positive scores initially. After the rebuttal, as most of the concerns have been addressed, the reviewers will likely to increase their scores.

---

### Decision · Program_Chairs · 2026-01-26

Accept (Poster)